# Neuronal HSF-1 coordinates the propagation of fat desaturation across tissues to enable adaptation to high temperatures in *C. elegans*

Laetitia Chauve[1], Francesca Hodge[1], Sharlene Murdoch[1], Fatemeh Masoudzadeh[1], Harry-Jack Mann[2], Andrea F. Lopez-Clavijo[2], Hanneke Okkenhaug[2], Greg West[2], Bebiana C. Sousa[2], Anne Segonds-Pichon[2], Cheryl Li[1], Steven W. Wingett[2], Hermine Kienberger[3], Karin Kleigrewe[3], Mario de Bono[4], Michael J. O. Wakelam[5†], Olivia Casanueva[1]*

1 Epigenetics Department, Babraham Institute, Cambridge, United Kingdom, 2 Babraham Institute, Cambridge, United Kingdom, 3 Bavarian Centre for Biomolecular Mass Spectrometry, Freising, Germany, 4 Institute of Science and Technology, Klosterneuburg, Austria, 5 Signalling Department, Babraham Institute, Cambridge, United Kingdom

† Deceased.
* moc771@gmail.com

**Data Availability Statement:** The RNA-seq data generated in this study has been deposited at NCBI under the GEO accession numer GSE145123. All

## Abstract

To survive elevated temperatures, ectotherms adjust the fluidity of membranes by fine-tuning lipid desaturation levels in a process previously described to be cell autonomous. We have discovered that, in *Caenorhabditis elegans*, neuronal heat shock factor 1 (HSF-1), the conserved master regulator of the heat shock response (HSR), causes extensive fat remodeling in peripheral tissues. These changes include a decrease in fat desaturase and acid lipase expression in the intestine and a global shift in the saturation levels of plasma membrane's phospholipids. The observed remodeling of plasma membrane is in line with ectothermic adaptive responses and gives worms a cumulative advantage to warm temperatures. We have determined that at least 6 TAX-2/TAX-4 cyclic guanosine monophosphate (cGMP) gated channel expressing sensory neurons, and transforming growth factor ß (TGF-β)/bone morphogenetic protein (BMP) are required for signaling across tissues to modulate fat desaturation. We also find neuronal *hsf-1* is not only sufficient but also partially necessary to control the fat remodeling response and for survival at warm temperatures. This is the first study to show that a thermostat-based mechanism can cell nonautonomously coordinate membrane saturation and composition across tissues in a multicellular animal.

## Introduction

Adaptation to high temperatures is fundamental for survival. The model organism *Caenorhabditis elegans* survives and reproduces over an environmental temperature range of 12˚C to 25˚C with an optimal temperature of 20˚C [1]. When exposed to heat stress, *C. elegans* activate a highly conserved stress response, the heat shock response (HSR), during which the heat

data and supplementary tables can be found here: https://zenodo.org/record/5524609#.YUx0eS1Q3tE.

**Funding:** OC was founded by grant ERC 638426 from European Research Council (https://erc.europa.eu/) and by Biotechnology and Biological Sciences Research Council [BBS/E/B000C0426] (https://bbsrc.ukri.org/). The funders had no role in study design, data collection and analysis, decision to publish, or preparation of the manuscript.

**Competing interests:** The authors have declared that no competing interests exist.

**Abbreviations:** BMP, bone morphogenetic protein; cGMP, cyclic guanosine monophosphate; DE, differentially expressed; Ds RNA, double-stranded RNA; EV, empty vector; *Ex hsf-1neuro*, extrachromosomal neuronal overexpression of *hsf-1*; FA, fatty acid; GO, gene ontology; GC–MS, gas chromatography coupled to mass spectrometry; HSF-1, heat shock factor 1; *hsf-1neuro*, neuronal overexpression of *hsf-1*; HSP, heat shock protein; HSR, heat shock response; HVA, homeoviscous adaptation; KD, knockdown; LD, lipid droplet; Lof, loss of function; OA, oleic acid; PC, phosphatidylcholine; PE, phosphatidylethanolamine; PI, phosphatidylinositol; PL, glycerophospholipid; qRT-PCR, quantitative RT-PCR; RNAi, RNA interference; RNA-seq, RNA sequencing; TGF-β, transforming growth factor ß; VNC, ventral nerve cord; WT, wild-type.

shock factor 1 (HSF-1) transcription factor rapidly induces the expression of heat shock proteins (HSPs) [2] involved in the refolding or clearance of heat-damaged proteins [3]. In addition to such cell-autonomous responses, the activation of stress responses in the nervous system helps to integrate external and internal cues across tissues to regain homeostasis; this is the case for heat shock and other stresses [4,5].

Neurons are crucial in regulating many physiological processes cell nonautonomously, including systemic accumulation of fat and the rebalance of energy. Across species, there are many well-established examples of neuronal circuits, involving a variety of neurotransmitters and neurohormonal signals, which modulate fat metabolism [6–9]. In *C. elegans*, many neurohormonal signals converge on the regulation of fat stores in the gut by controlling either the expression of catabolic lipases [10] or of fat desaturases involved in de novo fatty acid (FA) synthesis [11–13]. Notably, the transforming growth factor ß/bone morphogenetic protein (TGF-β/BMP) signaling pathway provides a good example of this regulation. DBL-1 is the *C. elegans* sole TGF-β/BMP ligand, homologous to vertebrate BMP10 [14,15]. DBL-1 is secreted from neuronal cells to control body size, reproductive output, and fat stores [12]. Both the environmental triggers that control the release of neurohormonal signals, such as DBL-1, and the adaptive value of this form of regulation remain unexplored.

In this study, neuronal overexpression of *hsf-1* (*hsf-1neuro*) was used as a tool to study the systemic consequences of neuronal *hsf-1* activation. We found that *hsf-1neuro* decreased the expression of fat desaturases *fat-6/fat-7*—orthologues of the human stearoyl-CoA desaturase—while activating the expression of catabolic lysosomal lipases, with consequent depletion of fat stores in the intestine. We find that the quantity and composition of phosphatidylinositol (PI) and phosphatidylethanolamine (PE)—phospholipids that make the bulk of the plasma membrane—are changed, and a notable shift in the saturation levels of the FA composition of the plasma membrane is observed. These changes are in line with ectothermic adaptive responses that rely on correcting the changes that temperature imposes on the fluidity of the plasma membrane [16,17]. In homeoviscous warm adaptation, membrane bilayers undergo a reversible change toward an ordered structure by increasing FA saturation [18–20], and an increase in the saturation of the FAs has a corrective effect. Consistently, we find that *hsf-1neuro* animals have a cumulative health advantage at warmer temperatures.

Homeoviscous adaptation was first discovered in single-celled organisms in which cell-autonomous sensors embedded in membranes control its fluidity [21]. In *C. elegans*, the heat-induced Acyl-CoA dehydrogenase, ACDH-11 [22] plays a similar cell-autonomous role. Here, we show that *hsf-1neuro* worms raised at the permissive temperature of 20°C constitutively activate a fat remodeling transcriptional program similar to that seen in the wild-type (WT) strain grown at 25°C. Our findings show that the cyclic guanosine monophosphate (cGMP) gated cation channel TAX-2/TAX-4 expressed in at least 6 sensory neurons and the suppression of TGF-β/BMP signaling are essential for the communication of neuronal *hsf-1* activity.

We propose that the exquisite sensitivity of neurons to external temperature could operate as a thermostat to control a fat remodeling program geared toward the physiological reoptimization of peripheral tissues to warmer temperatures. Consistent with this model, we observe that head neurons in WT animals are the only tissue that show increased activation of *hsf-1*–dependent gene expression at 25°C and that the activity of *hsf-1* is necessary to non–cell autonomously coordinate acid lipase expression and to ensure long-term survival at 25°C. This is the first study to report that ectotherms might use thermostat-based mechanisms to centrally and cell nonautonomously coordinate complex adaptive responses to warming temperatures, not unlike mechanisms engrained in endotherms.

## Results

### Overexpression of *hsf-1* in neurons leads to a cumulative adaptive advantage at warmer temperatures, and it is sufficient to fine-tune desaturation of glycerophospholipids (PLs)

Several pieces of evidence indicate that HSF-1 functions to retard the aging process. First, *hsf-1* knockdown (KD) by RNA interference (RNAi) or using the *hsf-1 (sy441)* allele that lacks the carboxyl-terminal transactivation domain causes animals to age faster [23–26], while an opposite phenotype is found upon ubiquitous *hsf-1* expression [23,24]. The fact that neurons are key to HSF-1–dependent life span modulation has been evidenced by the strong extension caused by the pan-neuronal expression of *hsf-1^neuro^* [5]. All these observations have been gathered at the standard cultivation temperature of 20˚C; however, an unexplained temperature dependency has been evidenced by the inability of either RNAi KD or of *hsf-1 (sy441)* to alter life span at 15˚C [26]. Consistently with this early observation, we noticed that *hsf-1^neuro^* life span extension was entirely dependent on growth temperature. We find that *hsf-1^neuro^* animals raised at 25˚C have twice lower the risk of death compared to WT; however, this protection is significantly reduced at 15˚C (based on hazard ratio WT/*hsf-1^neuro^* in **Fig 1A–1C** and **Table 1**). These observations are in line with a cumulative health advantage to warmer environments provided by *hsf-1^neuro^*.

One possibility is that *hsf-1^neuro^* animals are constitutively acclimated to warmer temperatures. If this was the case, then at least some of the transcriptional and metabolic responses that are mounted by animals raised at warmer temperatures should be shared by *hsf-1^neuro^* worms raised at lower temperatures. A key aspect of ectothermic temperature adaptation is that the increased fluidity of the membranes caused by warmer temperatures is counteracted by an increment in the saturation of the FA component of membrane phospholipids [16,17]. This effect is partially achieved by decreasing the expression and enzymatic output of stearoyl fat desaturases (encoded by *fat-7* and *fat-6* in worms) [19].

We reasoned that if *hsf-1^neuro^* animals were better suited to warmer temperatures, then the composition of membrane lipids should be similar to that of *fat-6;fat-7* mutants. To test this idea, we performed gas chromatography coupled to mass spectrometry (GC–MS) to quantify the levels of PLs that make up the bulk of the plasma membrane in animals raised at the standard 20˚C and their FA composition. To examine the acyl chain composition of the PLs, we determined the number of double bonds for FAs in position *sn-1* and *sn-2*. **Fig 1D** shows the specified number of double bonds (from 0 = saturated to 8 = polyunsaturated), and **Fig 1E** shows the desaturation index (defined in Methods [17]), as an example, we show PI 38 (PI with an acyl chain of 38 carbons). All genotypes phenocopy each other in that the number of double bonds decreases in mutants with respect to WT animals. **S2 Table** shows other examples of FAs with a reduced number of double bonds, primarily for PE and PI PL classes, which show a profile consistent with decreased fluidity. Acyl chain unsaturation is not the sole regulator of membrane fluidity, and in both insect and mammalian cells acclimated to warmer temperatures, there is a higher ratio of PE to phosphatidylcholine (PC) [17,27,28]. Since this ratio has not been previously quantified in nematodes, we measured the total amounts of PE and PC in WT animals grown at 15˚C, 20˚C, and 25˚C to find that consistent with reports in other species the ratio increases as a function of temperature (**Fig 1F**). *hsf-1^neuro^* animals—irrespective of their growth temperature—align with the ratios found in WT raised at 25˚C (**Fig 1F**). These results indicate that animals overexpressing *hsf-1^neuro^* phenocopy fat desaturase mutants by remodeling the composition of PLs consistent with membranes typical of warmer temperatures.

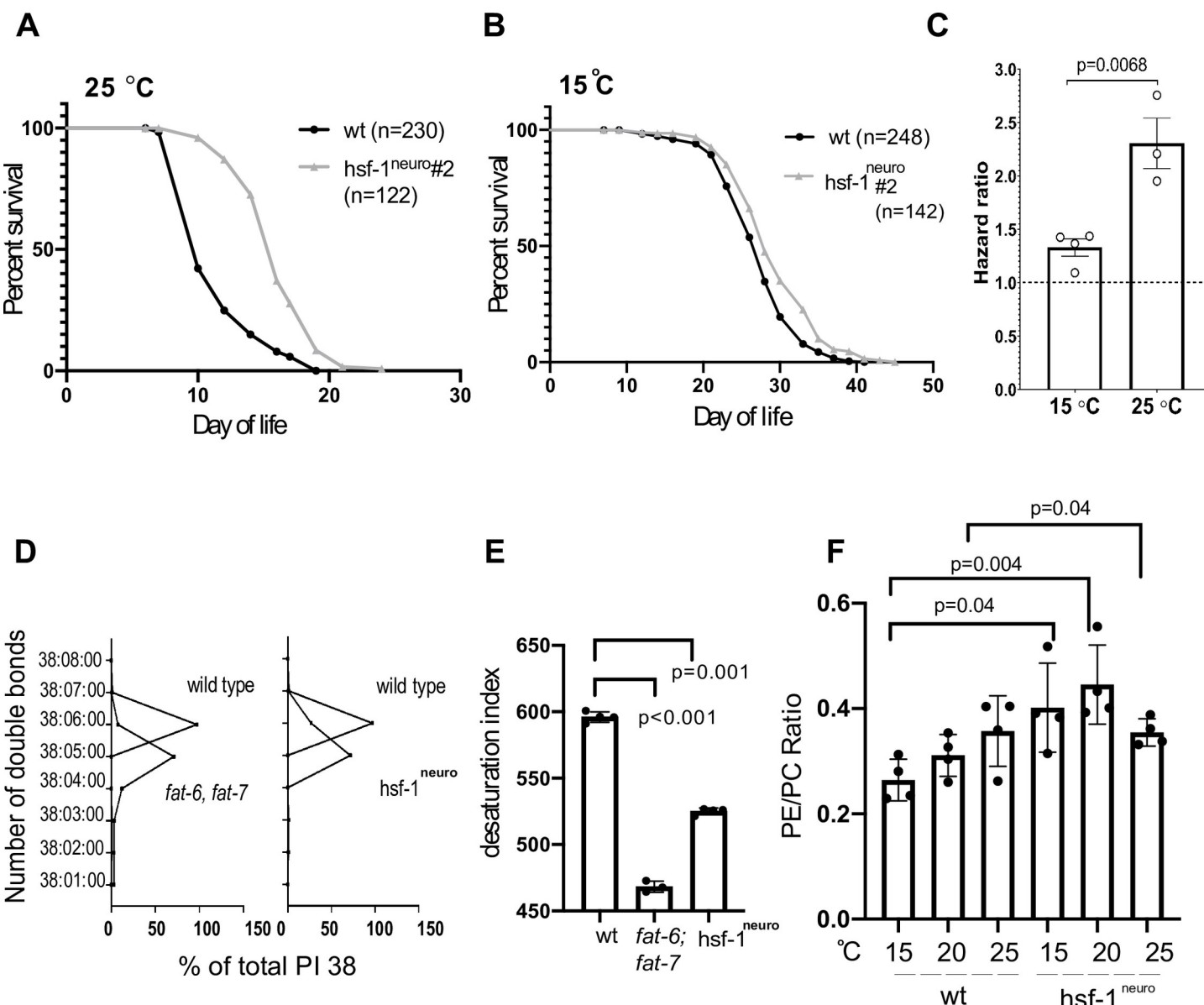

**Fig 1. A perceived warmer temperature alters the desaturation of membrane phospholipids in *hsf-1* worms, providing an adaptive advantage. (A–C)** *hsf-1neuro* is beneficial for life span at 25˚C but not at 15˚C. **(A)** Representative survival curve of WT (N2) and *hsf-1neuro #2 hsf-1neuro* line 2 (AGD1289) raised at 25˚C; wt: 230 deaths, 85 censored. *hsf-1neuro #2*: 122 deaths, 162 censored. Median survival wt: 10 days, *hsf-1neuro #2*: 16 days. Hazard ratio WT/*hsf-1neuro #2*: 2.756, *p*-value (log-rank test) WT versus *hsf-1neuro #2* <0.0001. **(B)** Representative survival curves of wt and *hsf-1neuro #2* raised at 15˚C. wt: 247 deaths, 244 censored. *hsf-1neuro #2*: 77 deaths, 418 censored. Median survival wt: 28 days, *hsf-1neuro #2*: 28 days. Hazard ratio WT/*hsf-1neuro #2*: 1.436. *p*-Value (log-rank test) wt versus *hsf-1neuro #2*: 0.0011. **(C)** When considering all biological replicates performed at 25˚C or 15˚C, the risk of death of WT is significantly higher at 25˚C, compared to *hsf-1neuro #2*. Histogram depicting the hazard ratio WT/*hsf-1neuro #2* for 4 biological replicates at 15˚C and 3 biological replicates at 25˚C. *p*-Value = 0.0068 (unpaired *t* test). All data and statistics for the life span experiments are listed in **Table 1**. **(D–F)** Phospholipids analysis using GC–MS shows changes in the lipid composition of *hsf-1neuro* animals are consistent with decreased saturation and fluidity in the plasma membrane. **(D)** Relationship between the quantity (in ng/ng DNA) of acyl chain (carbon number related to the FA composition in position *sn-1* and *sn-2* in PLs) for the specified number of double bonds (from 0 = saturated to 8 = polyunsaturated) in PI 38 (PI with an acyl chain of 38 carbons) for wt (N2) versus *fat-6(tm331)*; *fat-7(wa36)* (BX156) (left) and for wt versus *hsf-1neuro #2* (AGD1289) (right). The statistical test used was 1-way ANOVA. The *x* axis represents the percentage with regard to the total amount of FA in each mutant. Normalized data and statistics in S1 Table. **(E)** Desaturation index where higher numbers correspond to a higher number of double bonds; therefore, a lower saturation level, 1-way ANOVA, was the statistical test used. All normalized data are listed in S2 Table. **(F)** Shows the ratio of the total amounts of PE to PC in both WT animals and *hsf-1neuro* grown at 15˚C, 20˚C, and 25˚C. All normalized data are listed in S1 Table. Across D–F graphs, each dot corresponds to a biological replicate, and the bars correspond to the SEM. All data are provided in https://zenodo.org/record/5547464#.YVw6hX28rIV, and Supporting information tables can be found in https://zenodo.org/record/5547464#.YVrj1cYo-3U. FA, fatty acid; GC–MS, gas chromatography coupled to mass spectrometry; HSF-1, heat shock factor 1; *hsf-1neuro*, neuronal overexpression of *hsf-1*; PC, phosphatidylcholine; PE, phosphatidylethanolamine; PI, phosphatidylinositol; PL, glycerophospholipid; SEM, standard error of the mean; WT, wild-type.

**Table 1. Life span measurements.**

| Strain | T˚C | BR# | # deaths | # censored | Median survival | p-Value (versus CTR) log-rank test | Hazard ratio CTR/*hsf-1^neuro* #2 | p-Value hazard ratio 25˚C versus 15˚C |
|---|---|---|---|---|---|---|---|---|
| N2 (CTR) | 25˚C | 1 | 230 | 85 | 10 | | | Unpaired *t* test |
| AGD1289 (*hsf-1^neuro* #2) | 25˚C | 1 | 122 | 162 | 16 | <0.0001 (****) | 2.76 | 0.0068 (**) |
| N2 (CTR) | 25˚C | 2 | 212 | 86 | 10 | | | |
| AGD1289 (*hsf-1^neuro* #2) | 25˚C | 2 | 181 | 121 | 15 | <0.0001 (****) | 2.21 | |
| N2 (CTR) | 25˚C | 3 | 332 | 59 | 14 | | | |
| AGD1289 (*hsf-1^neuro*#2) | 25˚C | 3 | 161 | 236 | 17 | <0.0001 (****) | 1.95 | |
| N2 (CTR) | 15˚C | 3 | 180 | 301 | 28 | | | |
| AGD1289 (*hsf-1^neuro*#2) | 15˚C | 3 | 120 | 344 | 28 | 0.3661 (ns) | 1.09 | |
| N2 (CTR) | 15˚C | 1 | 248 | 254 | 28 | | | |
| AGD1289 (*hsf-1^neuro*#2) | 15˚C | 1 | 142 | 360 | 28 | <0.0001 (****) | 1.44 | |
| N2 (CTR) | 15˚C | 4 | 247 | 244 | 26 | | | |
| AGD1289 (*hsf-1^neuro*#2) | 15˚C | 4 | 77 | 418 | 30 | 0.0011 (**) | 1.43 | |
| N2 (CTR) | 15˚C | 2 | 193 | 309 | 28 | | | |
| AGD1289 (*hsf-1^neuro*#2) | 15˚C | 2 | 100 | 401 | 28 | 0.0022 (**) | 1.37 | |

#, number; BR, biological replicate; CTR, control; *hsf-1^neuro*, neuronal overexpression of hsf-1.

## WT animals raised at warm temperatures and *hsf-1^neuro* raised at lower temperatures share a common transcriptional fat remodeling program

If *hsf-1^neuro* animals were constitutively acclimated to warmer temperatures, then they should share at least some of the transcriptional responses that animals mount in response to growth at 25˚C. To explore this point, we identified differentially expressed (DE) genes between *hsf-1^neuro* (20˚C) and wt (20˚C) by DEseq2 (see Methods) and found 2,136 DE genes (**S3 Table**). These data were compared to DE genes from published sources that compared the transcriptomes of wt animals grown at 15˚C with animals grown at 25˚C [29]. Using different filtering criteria, we observed an overlap significantly higher than expected by chance (**S4 Table**). The Venn diagram in **Fig 2A** shows that the 2 sets of DE genes overlap by about 10%. **Fig 2B** shows the gene ontology (GO) categories of the overlapping genes, including lysosomal lipases required for lipid degradation (highlighted in **Fig 2C**). *C. elegans* contain at least 5 lysosomal lipases, LIPL-1 to LIPL-5 [30,31]. Some lipases are activated by the acidic environment of the lysosome to break down lipid droplets (LDs) and recycle FAs back into the cytosol, where they can be broken down by β-oxidation or recycled into membranes [32]. We observed that the transcripts encoding *lipl-1/2/3* and *lipl-5* are up-regulated in *hsf-1^neuro* compared to WT animals raised at the same temperature (20˚C) (**Fig 2D**).

LIPL-1 and LIPL-3 belong to the family of adipocyte triglyceride lipase–like patatin domain–containing lipases and have been described as fasting-induced lipases [31]. To determine if the transcriptional induction of lipases in *hsf-1^neuro* is due to starvation, we monitored the expression of the starvation biomarker *mir-80* [33] as well as pharyngeal pumping rates, which when slowed can reduce or preclude a worm's feeding ability [34–36], and are also increased upon starvation [37]. However, both *mir-80* levels and pumping rates were similar

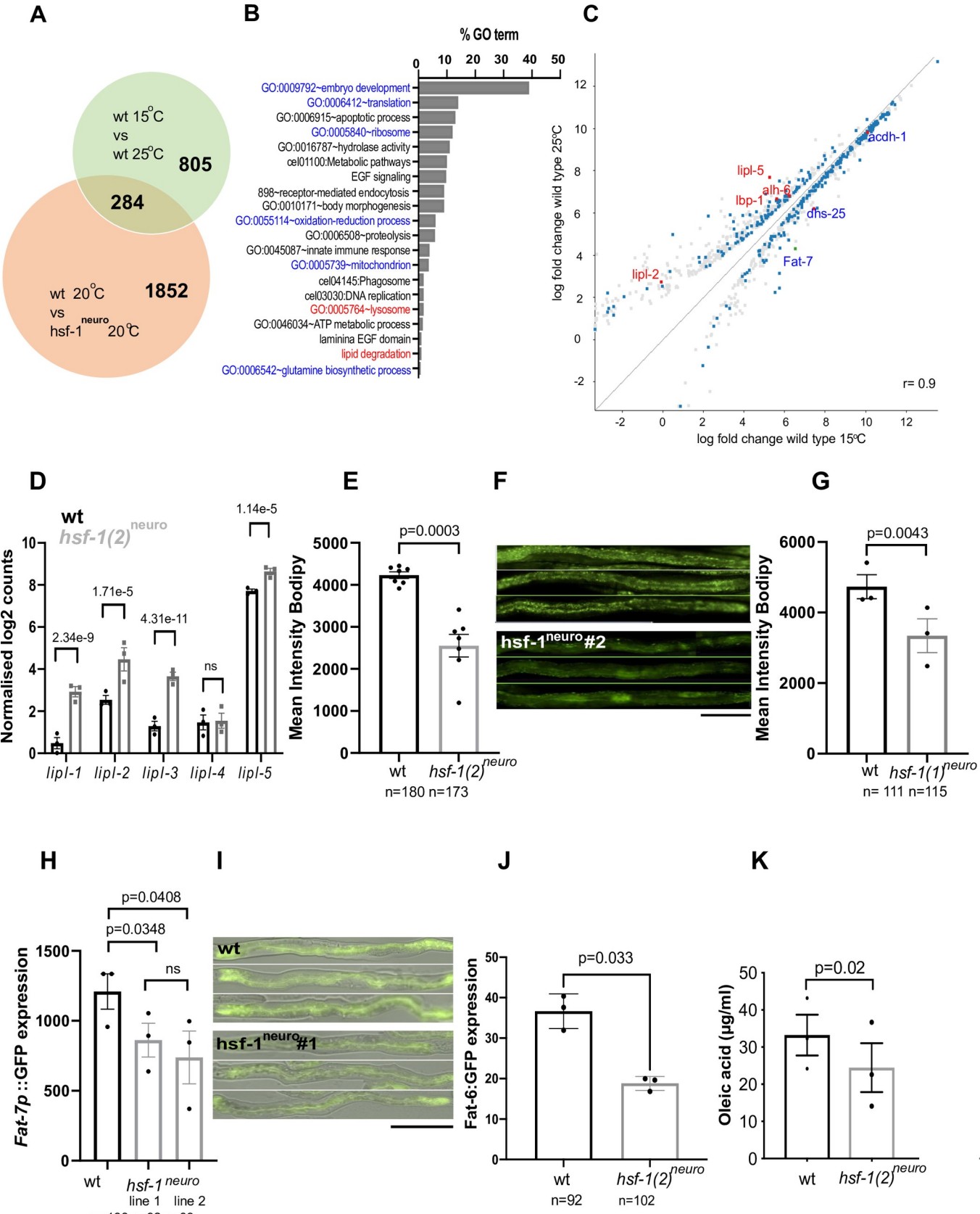

**Fig 2. Overexpression of *hsf-1neuro* in neurons activates a fat remodeling transcriptional program that resembles one activated by worms acclimated at 25°C. (A–C)** Warm temperature and *hsf-1neuro* share a transcriptional signature related to fat metabolism. **(A)** Transcriptomics analysis is described in the Methods section. The figure shows the overlap between genes DE between 25°C and 15°C (in the green circle). Raw data from [29] were reanalyzed using DESeq2 with a *p* < 0.05 cutoff, which gave a total of 1,089 genes, and genes DE between WT and *hsf-1neuro #2* (AGD1289, *hsf-1neuro* line 2) (in the orange circle) (Desq-2 using a cutoff *p* < 0.005, which gave 2,136 genes (**S3 Table**). A hypergeometric distribution was employed to determine the probability of overlap when using different filtering criteria (0, 0.05, 1, 1.5, or 2 log2-fold change). The overlap was found to be significantly different from chance using any of the mentioned filtering criteria (**S4 Table**). **(B)** Percentage of enrichment in GOs found using DAVID of the 284 genes that overlap between both experiments. GOs are highlighted if most genes within that class were down-regulated (blue) or up-regulated (red) in both *hsf-1neuro #2* and in worms grown at 25°C. **(C)** Scatter plot highlights the correlation (where x and y are shown in the same scale) of the quantified values (in log scale) for wt animals grown at 25°C versus 15°C. The probes highlighted correspond to the overlap between the 2 experiments (15°C versus 25°C and WT versus *hsf-1neuro #2*) for probes that are related to lipid metabolism. Red shows up-regulation at both 25°C and in *hsf-1neuro #2* and blue down-regulation. **(D)** Lipase mRNAs levels are increased in *hsf-1neuro #2* (AGD1289) compared to WT (N2). Normalized log counts obtained by transcriptomics analysis (described in Methods). *lipl-1*, *lipl-2*, *lipl-3*, and *lipl-5* mRNA levels are significantly increased in *hsf-1neuro #2* according to DESeq2. All DE genes are shown in **S3 Table**. **(E–G)** *hsf-1neuro* causes a depletion of fat stores. LD visualization using BODIPY to compare **(E)** *hsf-1neuro#2* (AGD1289, *hsf-1neuro* line 2) and **(G)** *hsf-1neuro #1* (MOC141, *hsf-1neuro* line 1) with WT (N2, wt). Mean intensity fluorescence levels are reduced by 29.3% and 40% in *hsf-1neuro #1* (mean intensity of 4731 (+/−479) versus 3343 (+/−340) and *#2*, (mean intensity of 4,231 (+/−79) versus 2,554 (+/−268), respectively, indicating that *hsf-1neuro* animals are leaner than WT controls. Statistics were performed using paired *t* test (**S5 Table**). **(F)** Images of straightened animals showing GFP fluorescence in the intestine from animals stained with BODIPY, at 20× magnification. **(H–K)** Neuronal *hsf-1neuro* overexpression decreases the expression of *fat-6/7* and their enzymatic output, OA **(H)** Average GFP fluorescence intensity driven by the *fat-7* promoter is decreased by 38.9% in *hsf-1neuro # 1* (MOC227, compare mean intensity of 1209(+/−126) versus 862(+/−121)) and 28.7% in *hsf-1neuro # 2* (MOC151, compare mean intensity of 1209(+/−126) versus 738(+/−189)) compared to *fat-7p::GFP* controls (HA1842). *p*-Values were calculated using a 1-way ANOVA (**S6 Table**). **(I)** Overlaid GFP fluorescence and DIC images taken at 20× magnification. **(J)** Average GFP fluorescence intensity driven by the *fat-6* promoter is decreased by 51% in *hsf-1neuro # 2* (MOC299) compared to controls (BX115, compare mean intensity of 36.6 +/− 2.5 versus 18.8 +/−0.99). *p*-Values were calculated using a paired *t* test (**S6 Table**). **(K)** *hsf-1neuro* causes a decrease in OA. GC–MS analysis was performed to quantify total FA levels (in μg per ml) in *hsf-1neuro #2* (AGD1289) compared to WT controls. OA (C18, c9) is decreased in *hsf-1neuro*. Normalized lipidomics data and statistics can be found in **S7 Table**. For all graphs, the SEM is shown; each dot represents a paired biological replicate, and *n* values indicate the number of worms used in total. Animals were raised at 20°C unless otherwise noted. Scale bar: 200 μm. All data are provided in https://zenodo.org/record/5547464#.YVw6hX28rIV, and Supporting information tables can be found in https://zenodo.org/record/5547464#.YVrj1cYo-3U. DE, differentially expressed; FA, fatty acid; GC–MS, gas chromatography coupled to mass spectrometry; GO, gene ontology; HSF-1, heat shock factor 1; *hsf-1neuro*, neuronal overexpression of *hsf-1*; LD, lipid droplet; OA, oleic acid; SEM, standard error of the mean; WT, wild-type.

in *hsf-1neuro* nematodes relative to wt (**S1A and S1B Fig**), ruling out starvation or an altered feeding behavior as explanatory mechanisms.

If activity of the lipases was increased, then their primary catabolic target, lipid stores, should be reduced. *C. elegans* do not have dedicated adipocytes instead storing fats in organelles called LDs in the intestine and hypodermis [32]. LDs are quantified using the lipid-intercalating fluorescent dye BODIPY [38]. We observed a reduction in BODIPY fluorescence of 29.3% in *hsf-1neuro #1* and 40% in *hsf-1neuro #2*, relative to wt young adult animals (**Fig 2E–2G**). Together, this evidence further confirms that a catabolic program activated by *hsf-1neuro* alters FA storage and distribution.

The fat desaturases FAT-6 and FAT-7 produce oleic acid (OA), which is an important component of LDs [19], and, therefore, a reduced activity may also contribute to decreased fat storages. *Fat-7* is also key to homeoviscous adaptation, and it is transcriptionally down-regulated in animals raised at 25°C compared to those raised at 15°C (**Fig 2C**) [19]. To study the effect of *hsf-1neuro* in vivo, we used a transcriptional *fat-7* reporter [39]. We find the fluorescent output of this reporter to be decreased in the 2 *hsf-1neuro* lines by 28.7% in *hsf-1neuro #1* and by 38.9% in *hsf-1neuro #2* relative to wt (**Fig 2H and 2I**). Similarly, an in vivo reporter of *fat-6* is decreased by 51% in *hsf-1neuro #2* worms compared to age-matched control animals (**Fig 2J**). If the observed decrease in the transcriptional output of these enzymes in *hsf-1neuro* worms was accompanied by a concomitant reduction of their enzymatic output, a reduction in OA levels is to be expected. To quantify free FA composition, we used GC–MS to measure total levels of free FA species. This analysis showed that, although the composition of most FAs remained stable in *hsf-1neuro*, the level of C18:1/OA is significantly reduced by 27% with respect to WT in *hsf-1neuro* animals compared to wt controls (**Fig 2K, S7 Table**). Together, these data further confirm that multiple aspects of lipid metabolism, including catabolism of FAs from lipid

stores, de novo synthesis of unsaturated FA, and production and saturation of membrane phospholipids, are altered in *hsf-1^neuro* animals.

## TAX-2/TAX-4 cGMP channels are essential for the fat remodeling phenotype caused by ectopic activation of neuronal stress

In *hsf-1^neuro* animals, *hsf-1* is expressed under the *rab-3* promoter that directs expression exclusively in all head neurons. To dissect the specific neurons that are responsible for the non–cell autonomous effect of *hsf-1^neuro* on fat metabolism, we first narrowed down neuronal classes by performing a suppressor RNAi screen. We selected 7 candidates based on their previously identified role on sensory neuron development or activity (Methods). As expected, when animals carrying an extrachromosomal array of *hsf-1^neuro* (*Exhsf-1^neuro*) were fed with an empty vector (EV), there is a significant 37% reduction in the transcriptional output of the *fat-7p*:: *GFP* compared to their wt siblings (**S6 Table**). We found that among the RNAi treatments, the loss of *tax*-2 caused the strongest suppression of the effect of extrachromosomal neuronal overexpression of *hsf-1* (Ex *hsf-1^neuro*) on *fat-7* expression (**Fig 3A**). TAX-2 and TAX-4 are the α and β subunits, respectively, of a heterodimeric cGMP gated cation channel that is required for the proper function of several sensory neurons linked to chemosensation, thermotaxis, and Dauer formation [40,41].

*Tax-4* RNAi had only a slight effect on *hsf-1^neuro*, and the difference with *tax-2* RNAi could be due to incomplete KD by the RNAi treatment, so we tested a loss-of-function (Lof) allele in the TAX-4 subunit. We find that *tax-4(p678)*, which causes dysfunction in all 12 TAX-2/4– expressing neurons [42], shows slightly higher levels of expression than controls; however, the difference does not reach significance (**Fig 3B and 3C**). As expected, the presence of *Ex hsf-1^neuro* significantly decreases the expression of *fat-7p*::*GFP* by 32.7% compared to wt. However, and consistent with the results of the *tax-2* RNAi KD, the Lof *tax-4(p678)* mutation suppressed this effect, by increasing *fat-7p*::*GFP* expression to 91.25% of WT levels (**Fig 3B and 3C**). As shown in **Fig 2**, *hsf-1^neuro*, in addition to decreasing *fat-6/7*, also up-regulates several lipases, including *lipl-1* by 11-fold (**Fig 3D**) and *lipl-3* by 6-fold (**Fig 3E**). We find a similar the effect on *lipl-1/3*, which are reverted to WT levels in the absence of functional cGMP activity (**Fig 3D and 3E**). If all the enzymatic activities are reverted to almost WT levels, then the expectation is that a Lof in the cGMP gated channel should also cause replenishment of fat stores in *hsf-1^neuro* animals. Consistent with this idea and as shown in **Fig 3F and 3G**, while a cGMP gated channel mutant did not significantly alter LD levels in WT worms, it completely rescues LD levels in *hsf-1^neuro*. In addition to *tax-2/4* effect on fat remodeling, we find that the loss of *tax-2* function also suppresses the life span extension conferred by *hsf-1^neuro*, further strengthening their functional link (**S2 Fig**). It is important to notice that despite the large rescue by both *tax-2* and *tax-4* mutants, the rescue is not complete (**Fig 1C**, **S2 Fig**), suggesting either incomplete Lof or the existence of additional modifiers. These results indicate that *hsf-1^neuro* cell nonautonomous effect on the enzymes that control LD homeostasis requires TAX-2/4– expressing neurons either directly or indirectly.

To further test if the activation of *hsf-1* acts directly in TAX-2/4–expressing neurons, we generated a transgenic line where *hsf-1* is expressed from the promoter of *tax-4* (**S3 Fig**). We find that in 2 independent overexpression lines, *hsf-1* overexpressed from *tax-4* promoter can recapitulate the effect of pan-neuronal *hsf-1* overexpression on the regulation of *fat-7p:gfp*, indicating that TAX-2/4–expressing neurons play a direct role for fat desaturase regulation. However, expression of *hsf-1* from *tax-4* promoter does not recapitulate the effect of pan-neuronal *hsf-1* for *lipase 1–3* expression, indicating that, although *tax-2/4* mutants have a strong impact on lipase regulation, this effect is indirect.

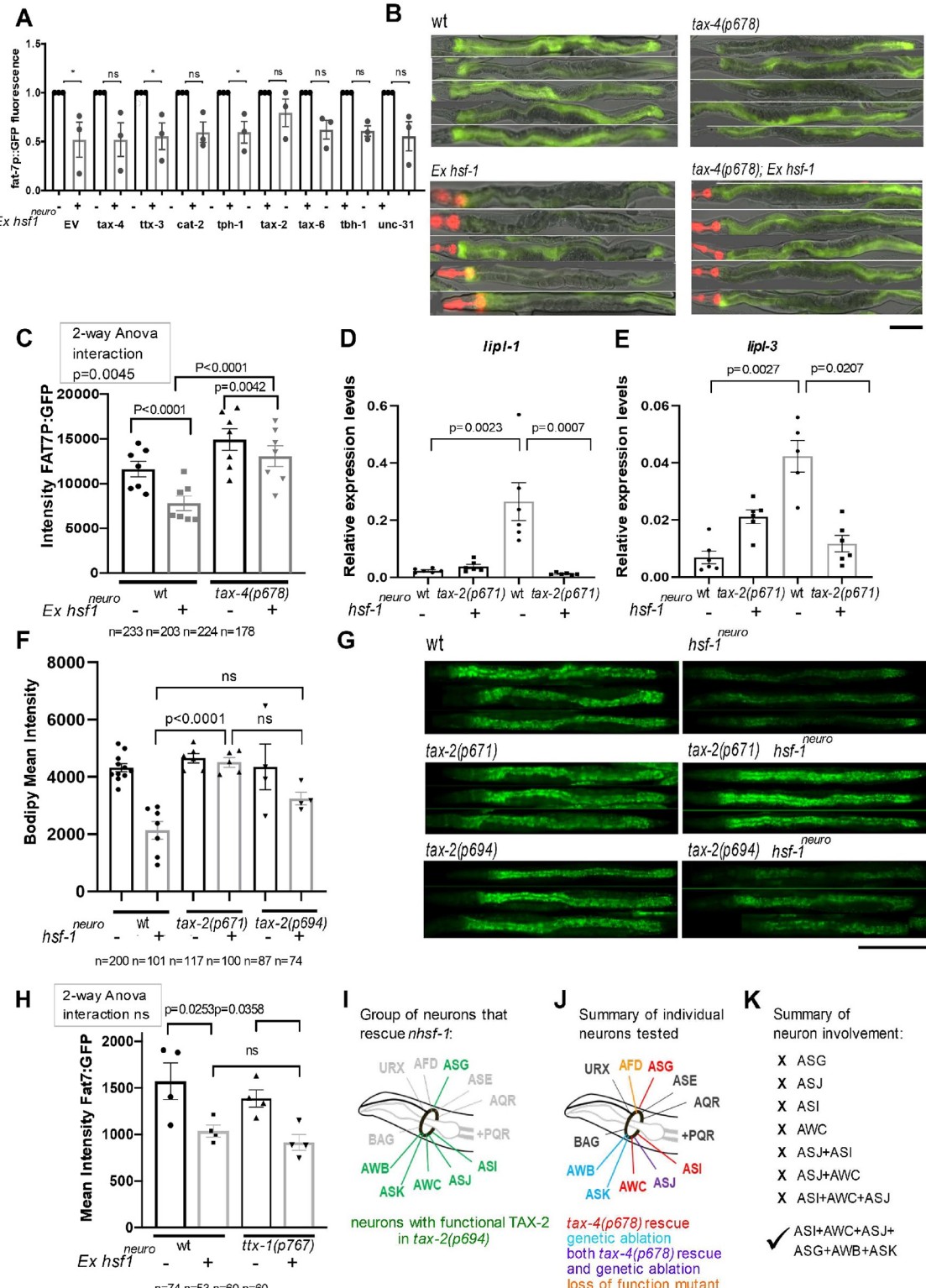

**Fig 3. Neuronal cells expressing the cGMP gated cation channel TAX-2/TAX-4 are required to modulate the effect of neuronal stress on fat-remodeling in the intestine. (A)** RNAi-based screen for suppressors of the repressive effect of *hsf-1neuro* on *fat-7* expression (details in Methods). *Ex hsf-1neuro; fat-7p::GFP; rrf-3* (MOC201) were treated with RNAi against 8 genes expressed in neuronal cells that are known or suspected to influence fat metabolism. The graph compares mean expression values of RNAi-fed *Ex hsf-1neuro* (+/−) pairs, where the average mean intensity values of GFP driven by *fat-7p::GFP* in animals containing the *Ex hsf-1neuro*

array were normalized to controls lacking the array. While most treatments do not change the effect of *Ex-hsf-1^neuro*, which causes a 37% drop in *fat7p*:*GFP* fluorescence, the treatment with *tax-2* RNAi is the strongest among the treatments, as it makes animals almost entirely refractive to the presence of *Ex hsf-1^neuro*, and the 19% difference among the *Ex hsf-1^neuro* pairs is not significant (all screen results in **S6 Table**). Because *tax-4* RNAi only has a moderate effect, we tested the effect of genetic mutants. **(B, C)** Epistasis experiment shows that *tax-4(p678)* and *Ex hsf-1^neuro* act in a linear pathway to decrease *fat-7*. Genetic epistasis using *Ex hsf-1^neuro;* *fat-7p::GFP* (MOC193) as control or crossed to a Lof mutant *tax-4(p678)* (MOC232). We used *tax-4(p678)* Lof mutation rather than *tax-2(p671)* mutation, as *tax-2* is on the same chromosome as *fat-7p::GFP* reporter. **(B)** 20× GFP fluorescence and bright field images of day 3 adult animals. **(C)** Average GFP intensity. Two-way ANOVA positive interaction indicates that the difference caused by the presence of *Ex hsf-1^neuro* is greater in control animals than in *tax-4(p678)*, corroborating suppression (**S6 Table**). **(D, E)** Relative expression levels of *lipl-1* and *lipl-3* mRNA by qRT-PCR in WT (N2), *tax-2(p671)* (PR671), *hsf-1^neuro #2* (AGD1289), and *tax-2(p671); hsf-1^neuro #2* (MOC252). The loss of *tax-2* function suppresses the increased levels of *lipl-1* and *lipl-3* in *hsf-1^neuro #2*. Statistics were performed using 1-way ANOVA for *lipl-1* and mixed-effect analysis for *lipl-3* (**S8 Table**). **(F–K)** The activation of *hsf-1^neuro* in 6 or more *tax-2/tax-4* expressing neurons is required for remote fat remodeling. **(F, H)** Average BODIPY fluorescence levels expression in **(F-G)** *tax-2(p671)* (PR671); *hsf-1^neuro #2* (AGD1289); *hsf-1^neuro #2; tax-2(p671)* (MOC252), *tax-2(p694)* (PR694) and *hsf-1^neuro #2;tax-2(p694)* (MOC293). **(H)** *ttx-1(p767))* (MOC297) and *Ex hsf-1^neuro, fat-7p::GFP* (MOC193). Across all graphs, SEM is shown; each dot represents a biological replicate, and *n* values indicate the number of worms used; *p*-values were obtained with a mixed effect analysis in F and 2-way ANOVA in H (**S9 Table**). **(G)** 20× fluorescence images showing BODIPY staining, scale bar: 200 μm. The *tax-2(p671)* null allele renders all *tax-2* expressing neurons non-functional and completely rescues the lean phenotype of *hsf-1^neuro*. This means that the activation of *hsf-1^neuro* in all or in a subset of TAX-2/4 sensory neurons can directly or indirectly cause fat redistribution. The *tax-2(p694)* allele is a partial Lof, where only the neurons colored in green (summarized in **Fig 3I**) remain functional. This allele causes a partial rescue (50%) of *hsf-1^neuro* fat stores. Therefore, all or some of the 6 remaining functional neurons, ASG, AWB, ASK, AWC, ASJ, and ASK, are required for *hsf-1^neuro*–dependent fat remodeling phenotype. **(I–J)** Diagrams showing TAX-2/4–expressing neurons. **(I)** Diagram summarizing the neurons that remain functional (green) in *tax-2 (p694)* partial Lof allele. **(J)** Diagram summarizing all the experiments performed to genetically dissect individual or combinations of neurons that mediate *hsf-1^neuro*–dependent fat remodeling (further explained in **S9 Table**). We used different techniques to test specific neuron(s) involvement in *hsf-1^neuro*–dependent lean phenotype, including Lof mutations (orange), *tax-4* rescue assays (red), genetic ablations (in blue), and both (in purple). K summarizes the results of all experiments described in **S9 Table**. All data are provided in https://zenodo.org/record/5547464#.YVw6hX28rIV, and Supporting information tables can be found in https://zenodo.org/record/5547464#.YVrj1cYo-3U. cGMP, cyclic guanosine monophosphate; *Ex hsf-1^neuro*, extrachromosomal neuronal overexpression of *hsf-1*; *hsf-1^neuro*, neuronal overexpression of *hsf-1*; Lof, loss of function; RNAi, RNA interference; SEM, standard error of the mean; WT, wild-type.

We sought to further narrow down specific TAX-2/4–expressing neurons that may be involved in this dual direct/indirect fat regulation. An obvious candidate is the AFD thermo-sensory neuron because it has been previously shown to act downstream of *hsf-1^neuro* animals to modulate HSR [5,43]. To test its involvement, we used mutations affecting the function of the AFD neuron (*ttx-1*) and its downstream neuron AIY (*ttx-3*) [44,45]; however, none of them is able to rescue *fat-7p::GFP* expression levels in *hsf-1^neuro* animals (**Fig 3H, S9 Table**). To further narrow down the identity of the relevant TAX-2/4–expressing neurons, we tested an allele with a deletion in the promoter region of *tax-2*, *tax-2(p694)*, which disrupts behaviors mediated by AQR, ASE, AFD, BAG, PQR, and URX neurons (**Fig 3I**) [40,46]. We find that this allele partially suppresses the fat accumulation phenotype of *hsf-1^neuro* (**Fig 3F and 3G, S9 Table**). These results indicate that neurons that are functionally affected by *tax-2(p694)* as well as neurons that are not affected by the *tax-2(p694)* allele are likely required by *hsf-1^neuro*. We focused our search on the group of neurons that is not altered by *tax-2(p694)*, namely ASG, ASJ, ASK, AWB, ASI, and AWC.

To narrow down the key neuron(s), we used 2 assays and measured 2 outputs for different neurons subsets (summarized in **Fig 3J** and **S9 Table**). First, we tested the effect of neuron-specific *tax-4* + rescue of *tax-4(p678)* null mutant, by measuring *fat-7p::GFP*. Second, we tested the effect of neuronal death (using neuron-specific expression of *caspase-1* or tetanus toxin) on lipase levels. Both assays were shown to be equivalent when used to either genetically ablate or rescue the ASJ neuron (**S9 Table**). We find that *tax-4(+)* expressed specifically in either ASJ, ASI, or AWC individually or in combinations (ASI and ASJ; AWC and ASJ; AWC and ASI; and ASI, ASJ, and AWC) do not rescue *fat-7p::GFP* expression (**S9 Table**). Second, we used genetic ablation to eliminate the function of AWB and ASK; however, eliminating their func-tion does not rebalance *lipl-1/3* levels in *hsf-1^neuro*. The results indicate that more than 6 TAX-

2/4–expressing neurons are responsible for remodeling and that they must be required for the regulation of a neuroendocrine signal(s) in order to cause cell nonautonomous alterations in lipid metabolism.

## Neuronal stress controls fat metabolism by decreasing TGF-β/BMP signaling

We next investigated how *hsf-1*[neuro] activity in neurons is signaled across tissues to remodel fat metabolism. When performing an in-depth phenotypic characterization of *hsf-1*[neuro] animals, we observed 2 phenotypes that indicate a potential candidate, which could be acting as a neuroendocrine signal. Firstly, *hsf-1*[neuro] animals were smaller in size relative to age-matched WT animals (**S4A Fig**) and have an extended reproductive period compared to WT animals (**S4B Fig**, **S10 Table**), suggesting a germline senescence phenotype. These phenotypes phenocopy the loss of the TGF-β/BMP signaling pathway [14,47]. The loss of DBL-1, the sole TGF-β/BMP ligand in *C. elegans*, also impacts *fat-6* and *fat-7* expression [12,47] and causes depletion of LDs [12,13], making it a promising candidate.

As overexpression of *hsf-1*[neuro] in neurons phenocopies a loss of TGF-β/BMP signaling, we tested the hypothesis that *hsf-1*[neuro] could work by suppressing TGF-β/BMP signaling. Because this pathway acts through SMAD transcription factors, we took advantage of a RAD-SMAD reporter, in which multiple, high-affinity SMAD-binding sites are placed upstream of GFP flanked with a nuclear localization signal, allowing a visual representation of TGF-β/BMP downstream transcriptional activation [48]. As shown in **Fig 4A–4C**, wt worms expressed GFP in both hypodermal nuclei (small <8 μm) and intestinal nuclei (large >8 μm) during the L4.8 stage. However, in the presence of *hsf-1*[neuro], the reporter's GFP signal is visibly reduced at the same developmental stage. As shown in **Fig 4B and 4C**, the RAD-SMAD signal decreases by 53% in large intestinal nuclei in *hsf-1*[neuro] #2 and by 28% in small hypodermal nuclei and by relative to controls. These results indicate that *hsf-1*[neuro] partially reduces the activity of the TGF-β/BMP pathway in multiple tissues.

To further test for this potential interaction, we performed genetic epistasis experiments. *Dbl-1* Lof mutants, such as *dbl-1(nk3)*, do not phenocopy *hsf-1*[neuro] animals with regard to *lipl-1* and *lipl-3* expression (**S8 Table**). However, and as described elsewhere [12,47], the loss of *dbl-1* function causes a reduction in *fat-7* expression, in line with our observations for *hsf-1*[neuro] (**Fig 4D and 4E**). Moreover, *dbl-1(nk3)* phenocopies *fat-6;fat-7* mutants and *hsf-1*[neuro] animals in PL density and composition (**S4C and S4D Fig**), and it is slightly leaner than WT animals, accumulating 25% less LDs than WT controls (**Fig 4F and 4G**) [12,13], but has an observable (although not statistically significant) larger amount of fat stores compared to *hsf-1*[neuro].

We hypothesized that if TGF-β/BMP signaling acts in the same linear pathway, downstream of *hsf-1*[neuro], then in the absence of TGF-β/BMP signaling, *dbl-1* should not be able to further reduce fat stores of *hsf-1*[neuro]. Consistent with this hypothesis, in animals carrying both *dbl-1(nk3)* and *hsf-1*[neuro] LDs levels are not significantly different relative to those observed in either *dbl-1(nk3)* or *hsf-1*[neuro] alone (**Fig 4F and 4G**). Together, these results indicate that the loss of LD accumulation in *hsf-1*[neuro] is caused, at least in part, by the loss of TGF-β/BMP signaling pathway activity. In further support for a linear epistatic relationship between *hsf-1*[neuro] and *dbl-1(nk3)*, we observed that the *dbl-1(nk-3); hsf-1*[neuro] animals are not shorter than either single mutant (**S4A Fig**) and that the decrease in *fat-7p*::*GFP* fluorescence is not further decreased in *dbl-1(k3); hsf-1*[neuro] compared to either *hsf-1*[neuro] or *dbl-1(nk3)* alone (**Fig 4D**). To test the linear relationship further, we hypothesized that the overexpression of *dbl-1* should rescue fat remodeling in *hsf-1*[neuro]. However, overexpression of *dbl-1*—primarily in ventral nerve cord

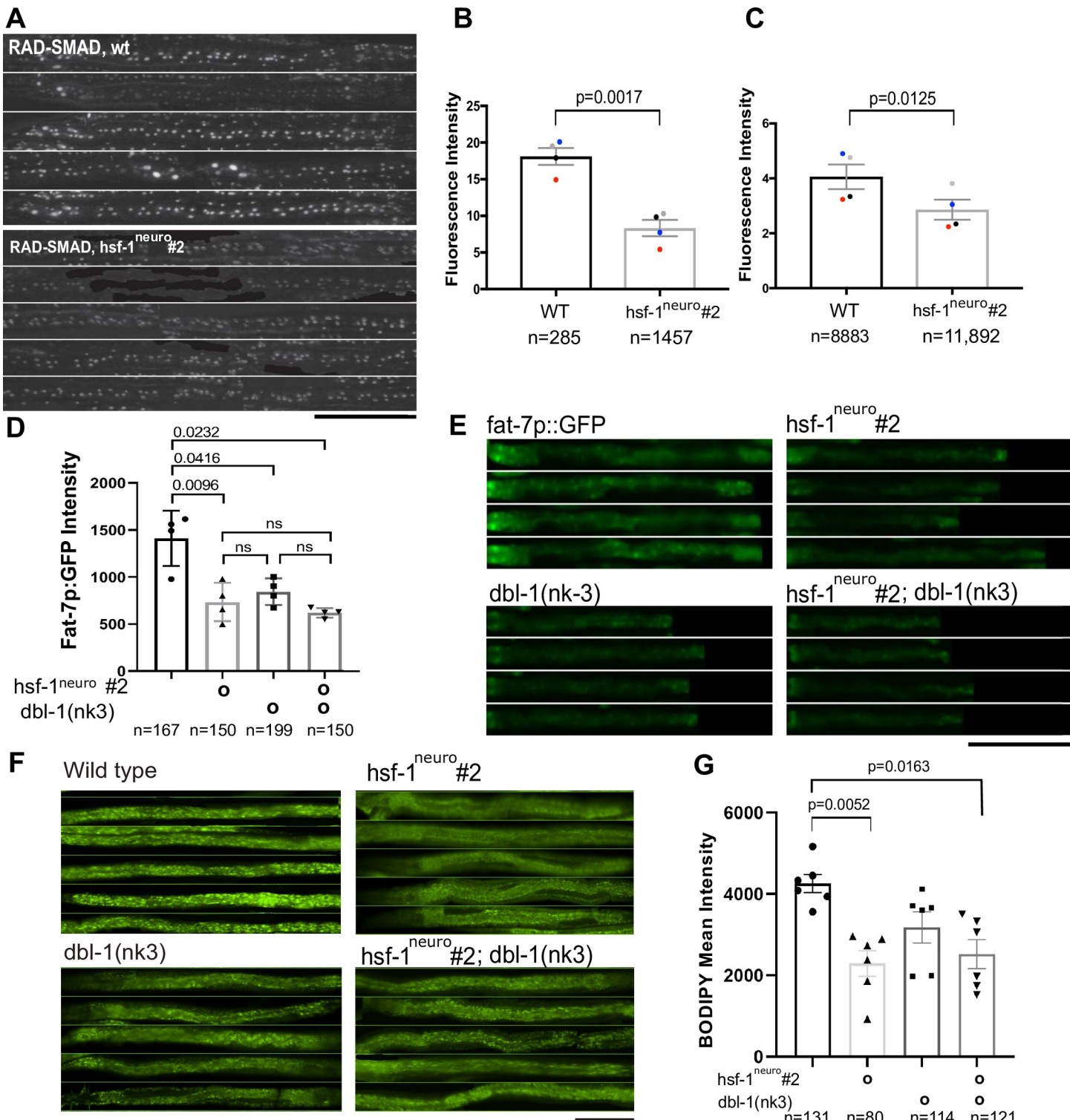

**Fig 4. Neuronal stress remodels fat metabolism by suppressing TGF-β/BMP signaling. (A–C)** Neuronal stress decreases the fluorescent GFP output of an in vivo RAD-SMAD reporter that directly and positively responds to TGF-β/BMP signaling [48]. **(A)** At 20°C, there is a reduction in nuclear GFP signal of L4.8-stage, *hsf-1neuro #2* animals (MOC229, lower panel) compared to age-matched controls (LW2436, upper panel). **(B)** Quantification of fluorescence levels of RAD-SMAD reporter in large intestinal nuclei (>60 um) **(C)** and in small hypodermal nuclei (<60 um). Paired *t* test shows a significant difference in the mean fluorescence levels of controls relative to *hsf-1neuro #2* animals (**S6 Table**). *N* values correspond to the number of fluorescent nuclei across all worms from 4 independent replicates (total number of worms: 131 in WT and 115 in*hsf-1neuro*). Paired replicates have been color coded to show consistency. **(D, E)** Genetic epistasis shows that *hsf-1neuro* and *dbl-1* work in a linear pathway to

decrease *fat-7* expression. Quantification of *fat-7p*::*GFP* fluorescence in age-matched L4.9 animals from the following backgrounds: WT (HA1842), *hsf-1neuro #2* (MOC151), and *dbl-1(nk3)* (MOC279) *hsf-1neuro #2;dbl-1(nk3)* (MOC347). Statistics were performed using a 1-way ANOVA (**S6 Table**). **(E)** GFP fluorescence images taken at 20× magnification. **(F, G)** Genetic epistasis suggests that *hsf-1neuro* and *dbl-1* work in a linear pathway to decrease fat stores. LDs were quantified using BODIPY staining in: *dbl-1(nk-3)* (NU3); *hsf-1neuro #2* (AGD1289); *dbl-1(nk3)*; *hsf-1neuro #2* (MOC254). **(F)** Representative images of BODIPY fluorescence images taken at 20× magnification. **(G)** Average intensity level of BODIPY fluorescence per biological replicate. *p*-Values were obtained by 1-way ANOVA (**S5 Table**). Across graphs: All experiments were performed with *hsf-1neuro* line 2 animals, error bars: SEM, each dot represents an independent biological replicate, *N* values correspond to number of worms, except in **Fig 4B and 4C** where it corresponds to nuclei. Scale bar: 200 μm. All data are provided in https://zenodo.org/record/5547464#.YVw6hX28rIV, and Supporting information tables can be found in https://zenodo.org/record/5547464#.YVrj1cYo-3U. BMP, bone morphogenetic protein; *hsf-1neuro*, neuronal overexpression of *hsf-1*; LD, lipid droplet; SEM, standard error of the mean; TGF-β, transforming growth factor ß; WT, wild-type.

(VNC) cells—using *dbl-1p*::*dbl-1*::*TM*::*mcherry* [49] behaves in the same manner as the loss of *dbl-1* function for *fat-7p*::*GFP* levels or fat accumulation [13] (**S5E Fig**). Although these results are surprising, they suggest that the exact dose of TGF-β/BMP expression is key.

Collectively, these data indicate that increased activity of *hsf-1neuro* in neurons suppresses TGF-β/BMP signaling, causing multiple cell nonautonomous changes including a reduction of *fat-7* expression, fat stores, and body size and that the exact dose of BMP is key to ensure appropriate fat remodeling. One potential mechanism is that *hsf-1neuro* changes the expression of *dbl-1* and of downstream signal transduction components of the SMA/BMP pathway. We have found that this is indeed the case, as the transcripts of *dbl-1*; *sma-3(R-SMAD)*; *sma-4 (Co-SMAD)* and *sma-6*—a type I receptor of TGF-β/BMP signaling—are all down-regulated when *hsf-1* activity is increased in neurons (**S5A Fig**). A reduction in transcripts could be caused by reduced transcriptional output or decreased transcript stability. The study of a transcriptional reporter of *dbl-1*—primarily expressed primarily from VNC cells [15]—suggest that this regulation may be posttranscriptional (**S5C and S5D Fig**).

Together, these experiments support a model, summarized in **Fig 5**, where *hsf-1neuro* overexpression in TAX-2/4–expressing sensory neurons decreases the TGF-β/BMP signaling pathway by altering the expression of multiple signaling components, subsequently causing membrane remodeling via down-regulation of *fat-6/7*. Other signals and neurons may be required to control the fat remodeling program controlled by acid lipases.

## Evidence for an HSF-1–dependent thermostat in WT animals grown at 25°C

Collectively, the data presented are consistent with the idea that the constitutive activation of *hsf-1* in sensory neurons causes animals to perceive a higher temperature than the one they are actually experiencing. It suggests that WT animals may use HSF-1 as a dose-sensitive system where neurons need activation of HSF-1 in order to activate a non–cell autonomous fat-remodeling program. If HSF-1 was used as a temperature sensor to tune external temperature with the appropriate fat metabolism, we would expect that in WT animals, neuronal cells induce a mild stress response in a temperature-sensitive manner. To test this model, we took advantage of cGAL, a temperature-robust and highly sensitive GAL4-UAS binary expression system optimized for *C. elegan*s [50] to monitor the expression of *hsps* at basal levels. As shown in **Fig 6A and 6B**, when GAL4 is driven by an HSF-1–dependent promoter (*hsp-16.41*), GFP expression is primarily restricted to cells that have axonal projections in the head region. This is consistent with spatial transcriptomics data [51] where all studied *hsp* transcripts are found in the head region of animals grown at 20°C and with single-cell transcriptomics where *hsps* are expressed in multiple neuron [52]. *Hsp* genes are known to be more highly expressed at 25°C relative to their expression levels below 21°C [20,26,29]. Consistent with this, we observed that the number of GFP-positive neurons increased 16-fold from 15°C to 25°C (**Fig 6C and 6D, S11 Table**). This interaction is not influenced by the cGAL system, because when

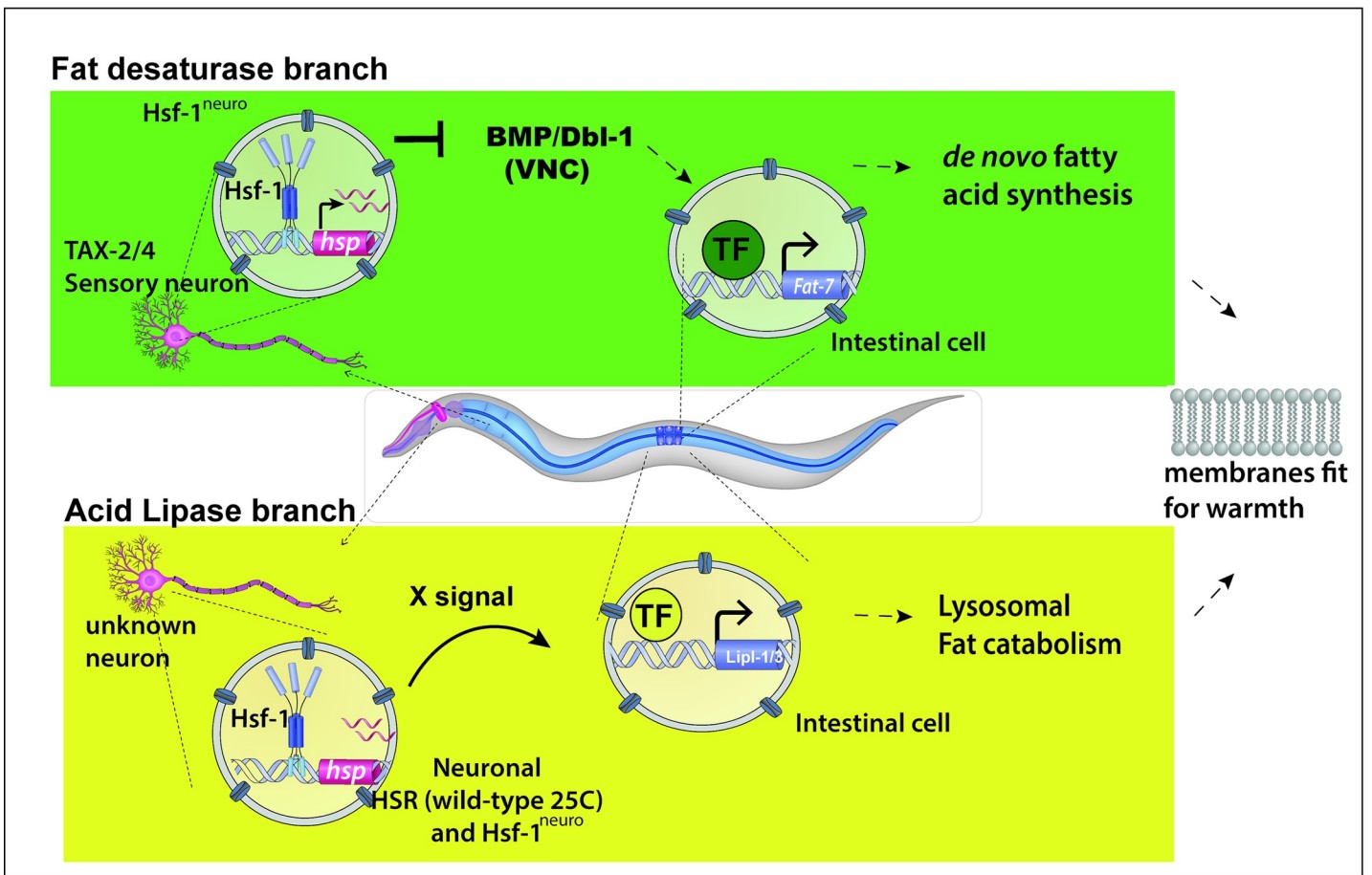

**Fig 5. Diagram depicting the 2 branches of the fat remodeling program controlled by neuronal activation of Hsf-1.** This diagram shows the image of an animal that has the "thermostat" turned on by *hsf-1^neuro* (in both branches) or by growing animals at 25˚C (only in the acid lipase branch). **Fat desaturase branch:** TAX-2/4–expressing neurons are sufficient but not necessary to control *fat-6/7* expression in the gut. The neurohormonal TGF-β/BMP signal is down-regulated in the VNC neurons. In this model, the correct dose of BMP is key for *Fat-6/7* expression in the gut and requires additional and redundant signals for *fat-6/7* expression. **Acid-lipase branch:** HSF-1 is both necessary and sufficient to control this pathway. The HSF-1 thermostat works independently of TGF-β/BMP signaling pathway. The HSR is not required within TAX-2/4–expressing neurons, but presumably in other neurons that are in close proximity to them. Both the neurohormonal signals and TFs required for this pathway remain unknown. Both pathways are required for fat remodeling and survival at warmer temperatures. BMP, bone morphogenetic protein; HSF-1, heat shock factor 1; *hsf-1^neuro*, neuronal overexpression of *hsf-1*; HSR, heat shock response; TF, transcription factor; VNC, ventral nerve cord.

*gfp* is driven by *unc-47*, a promoter that is restricted to GABAergic neurons and that lack HSF-1 binding sites, *gfp* expression is independent of temperature (**Fig 6E**).

One potential scenario is that the neurons that activate the HSR at 25˚C are TAX-2/4–expressing neurons. To determine if this was the case, we performed a co-localization experiment using 2 independent reporters (either mORANGE or wrmSCARLET driven by the *tax-4* promoter) in addition to the cGAL system to monitor neurons with an active HSR. As shown in **S6A and S6B Fig**, the analysis of both reporters indicates no direct co-localization between TAX-4 neurons and neurons with an active HSR. However, neuronal circuits are intricate and in close proximity. The careful analysis of mORANGE expression revealed, however, that co-localization can be occasionally observed. These observations are consistent with the idea that TAX-2/4 neurons play 2 distinct roles in the modulation of *hsf-1^neuro* phenotypes: a direct role to control *fat-7* via TGF-β/BMP signaling and an indirect role to control lipase expression. Additional support for the dual role is evidenced by the partial effect of *hsf-1* expressed from a *tax-4* promoter (**S2 Fig**) and by a triple epistasis experiment. In this experiment, we removed

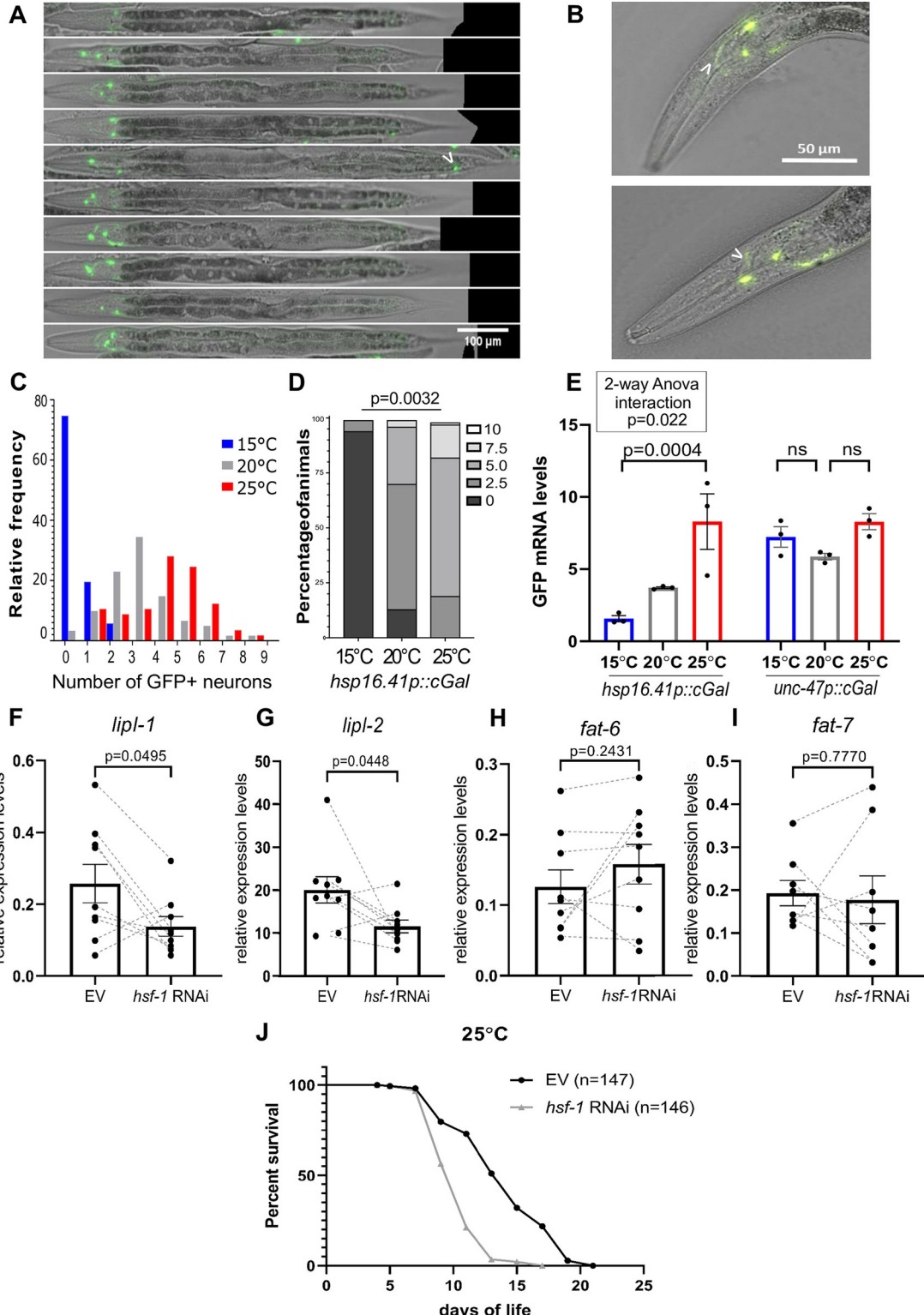

**Fig 6. Evidence for an HSF-1–dependent thermostat in WT animals. (A, B)** The figures show representative images taken at 20× magnification with both GFP fluorescence and Nomarski optics of young adult *hsp16.41p*:cGAL/UAS:GFP (PS7171) animals raised at 20°C. In the absence of stress, GFP is expressed exclusively in anterior cells and occasionally in tail neurons (arrowhead in A). The cells are neurons as they show axonal projections (in B, arrowhead points at a visible axonal projection). Animals shown were straightened using the Worm-align FIJI-based pipeline [53]. Scale bar: 100 μm (A), 50 μm (B). **(C)** Neuronal

expression of an HSF-1–dependent promoter is temperature sensitive. (C, D) The histogram shows the density of animals harboring 0 to 9 GFP expressing neurons at 15°C (blue), 20°C (gray), and 25°C (red) in *hsp16.41p:cGal/UAS:gfp* (PS7171). The number of GFP-positive neurons increases by 6.6-fold from 15°C to 20°C; 16-fold from 15°C to 25°C; and 2.5-fold from 15°C to 20°C. (D) Stacked bar showing the density of animals expressing 0, 2.5, 5, 7.5, and 10 GFP + neurons. Statistics were performed using a 1-way ANOVA test and provided in S11 Table. (E) *hsp16.41* temperature dependency is not caused by temperature sensitivity of the cGAL system. A cGAL system driven by the promoter of *hsp16.41* (PS7171) was compared to cGAL driven by the promoter of *unc-47* (PS7167), which is not responsive to HSF-1 qRT-PCR was used to obtain *gfp* mRNA levels in animals raised at 15°C (blue), 20°C (gray), and 25°C (red). A 2-way ANOVA analysis shows a positive interaction (*p* = 0.022) between temperature and genotype, indicating that the difference in *gfp* levels between 15°C and 25°C is higher for the promoter *hsp16.41* than for *unc-47* (S8 Table). (F–I) KD of *hsf-1* in neurons decreases expression levels of lipases but does not affect *fat-6* and *fat-7* mRNA expression levels. Relative expression levels of *lipl-1* (F), *lipl-2* (G), *fat-6* (H), and *fat-7* (I) mRNA by qRT-PCR in TU3401 RNAi sensitive genetic background. TU3401 animals were raised at 25°C for 1 generation and were fed either L4440 (EV) control or *hsf-1* Ds RNA (Ahringer RNAi library) for 48 hours, from late L4 onward. Levels of mRNA levels were measured at day 3 of adulthood. Statistics were performed using paired *t* test (S8 Table). To control for a potential spurious effect due to *unc-119p* leakiness into the epidermis, the same treatment was used on an epidermis-specific RNAi strain (QK52) (S8 Fig). While RNAi treatment was effective using both strains (Methods), the only significant effect on lipase expression was observed in TU3401, indicating that the results presented are neuron-specific (S8 Table). Bars represent the SEM, and each dot represents a biological replicate obtained from either young adult animals (C, E) or day 3 animals (F–I). Scale bar: 200 μm. (J) Knocking down *hsf-1* in neurons reduces life span at 25°C. Representative survival curve of TU3401 animals (neuron-specific RNAi sensitive) fed either L4440 (EV) control or *hsf-1* Ds RNA since L1 stage. EV: 147 deaths, 14 censored, median survival: 15 days. *hsf-1* RNAi: 146 deaths, 16 censored, median survival: 11 days. *p*-Value (log-rank test) EV versus *hsf-1* RNAi <0.0001(****). Data for all biological replicates are in S13 Table. All data are provided in https://zenodo.org/record/5547464#.YVw6hX28rIV, and Supporting information tables can be found in https://zenodo.org/record/5547464#.YVrj1cYo-3U. Ds RNA, double-stranded RNA; EV, empty vector; HSF-1, heat shock factor 1; KD, knockdown; qRT-PCR, quantitative RT-PCR; RNAi, RNA interference; SEM, standard error of the mean; WT, wild-type.

the function of both *tax-2* and *dbl-1* from *hsf-1^neuro* animals. If *tax-2* dependent suppression on fat accumulation was entirely dependent on counteracting the down-regulation of TGF-β/ BMP signaling by *hsf-1^neuro*, then removing the BMP ligand (*dbl-1*) should prevent *tax-2* suppression entirely. However, we find that *tax-2* can partially rescue the lean phenotype of *hsf-1^neuro* even in the absence of TGF-β/BMP signaling. The partial rescue is further support for the idea that 2 independent pathways—one of them not controlled by TGF-β/BMP—are influenced by the loss of *tax-2* (S6C Fig).

If the thermostat was playing a physiological role in WT animals, then the expectation is that removing *hsf-1* function in neurons should cause changes in the 25°C–dependent fat remodeling program of otherwise WT animals. To test this idea, we fed with *hsf-1* double-stranded RNA (Ds RNA), a strain of worms where RNAi is effective in neuronal cells. We find that a reduction of *hsf-1* function in neurons of worms grown for 1 generation at 25°C causes down-regulation of *lipl-1* and *lipl-2* mRNA expression compared to WT animals (Fig 6F and 6G), but it does not have an effect on *fat-6* or *fat-7* mRNA (Fig 6H and 6I). These results, together with observations made on *hsf-1^neuro*, indicate that neuronal HSF-1 is necessary and sufficient to cell nonautonomously control acid lipase expression in gut cells. However, HSF-1 is sufficient but not necessary to control *fat-6* and *fat-7* expression, suggesting that other redundant mechanisms are in place.

If the gene expression changes caused by neuronal *hsf-1* KD were functional, these should impair the survival of worms at warmer temperatures. We observe that this is the case because *hsf-1* KD in a strain where RNAi is effective primarily in neuronal cells shortens life span at 25°C compared to controls. The KD of *hsf-1* only changes lipase levels and does not change *fat-6/7* expression levels, so the reduced advantage at 25°C must be caused by decreased lipase levels in the gut. However, mutants lacking *fat-6* and *fat-7* (S7A Fig) or *dbl-1* function (S7B Fig) also shorten life span at 25°C, suggesting that both enzymatic branches controlled by *hsf-1^neuro* are key to survival at higher temperatures. However, this conclusion has to be drawn cautiously because, although we controlled for known spurious effects of the RNAi strain in

the hypodermis (**S8 Fig**), unreported leakage to other tissues may have an influence on the Lof phenotypes observed. We have summarized the model in **Fig 5**.

## Discussion

Our results show that the overexpression of *hsf-1* in neurons leads to LDs depletion and widespread remodeling of the abundance and composition of membrane PLs, in a manner that would support homeoviscous adaptation to high temperatures (**S9 Fig**). We identify 2 independent molecular pathways, the first regulates the fat desaturases *fat-6* and *fat-7*—with the latter normally down-regulated at 25˚C. The data presented are consistent with the idea that overexpression of *hsf-1* in 6 or more TAX-2/4 neurons is sufficient to switch off fat desaturation by decreasing TGF-β/BMP ligand availability and down-regulation of pathway component expression and activity. Additional signals are required, however, for this branch of the pathway to act under physiological conditions. The second branch is required for the up-regulation of acid lipases under warm temperatures, and it is controlled by unknown sensory neurons and signals. In the first *fat-6/7* branch, *hsf-1* expression in TAX-2/4 neurons is sufficient but not necessary to exert neuroendocrine control of fat remodeling, whereas in lipase branch, *hsf-1* is both necessary and sufficient and requires a key but indirect input from TAX-2/4 neurons (see model in **Fig 5**).

We currently know of 2 main strategies for dealing with temperature. Endotherms, such as mammals and avian species, maintain their body temperature in a process that involves the use of peripheral sensors to relay temperature information to the hypothalamus [54]. Here, specific neurons function as a type of thermostat that sets and maintains a specific body temperature. By contrast, ectotherms, such as reptiles, fish, and invertebrates, cannot control their own body temperature. In recent years, molecular studies have revealed that invertebrate model organisms, such as *Drosophila melanogaster* and *C. elegans*, use thermostat-based strategies to control temperature-seeking behaviors [55].

Behavioral adaptations alone are not enough to ensure their survival in ectotherms. This is because cell membranes, which consist primarily of PLs, are extremely sensitive to temperature. This raises the question: How do cell membranes adapt to temperature change? Work in bacteria, and subsequently in metazoans, has shown that a restricted, adaptive cell-autonomous mechanism exists that adjusts the cell membrane's FA content [22,56]. This mechanism involves sensors that capture information about membrane fluidity and correct it. For example, at cold temperatures, desaturated forms of FAs are preferred as they add entropy to membranes. Multicellularity, however, poses additional challenges because FAs are only produced in specialized fat storage tissues. The question then is how distant tissues get the desired FAs to construct their membranes?

Recently, work done using *C. elegans* has shed light onto cross-tissue adaptive responses to cold. First, the cold-sensor, PAQR-2, homologue of mammalian AdipoR2, can communicate information about membrane fluidity across tissues [57,58]. Secondly, a brain–gut axis has been shown to modulate cold habituation [59]. However, despite a cell-autonomous sensor aiding the nematode's membrane adaptation to heat [22], there is no information about how a multicellular system can adapt to warming temperatures.

From our findings, we propose that, although ectotherms cannot internally regulate their own body temperature, they may be centrally controlled similar to endotherms. Our results suggest that the conserved HSR, regulated by HSF-1, which normally responds to heat-induced protein misfolding, may have been co-opted by neurons to remodel fat metabolism across the organism's tissues. Although other studies have linked neuronal signals with fat metabolism [10,12], this is the first study to suggest that the heat-inducible response can

function in neurons to centrally coordinate membrane composition across tissues. Endotherms have peripheral thermostat sensors that relay information to the hypothalamus. We find that in worms, a "hypothalamus-like" group of neurons may be both sensing environmental temperature and coordinating a neuroendocrine response to adapt to it.

Our analysis indicates that when *hsf-1* is overexpressed in 6 or more TAX-2/TAX-4 functional neurons is sufficient to mediate the *fat-6/7* response. We observe that the HSR is activated only occasionally in TAX-2/4 neurons, but more often in other neurons that remain uncharacterized. We think that the neurons that activate the HSR constitute the actual "thermostat" and act in close collaboration to TAX-2/4 neurons to send additional signals to control fat desaturation. It remains to be understood what is the identity of these thermostat neurons and the mechanisms of action as to how they work in combination with TAX-2/4 neurons and what other neurohormonal signals do they control. Another question that remains open is how TAX-2/4–expressing neurons interact with HSR. We have observed that the HSR can be activated in neurons at 25°C even in the absence of *tax-2/tax-4* function, indicating that *tax-2/tax-4* mutations are not preventing the activation of the HSR in neurons, but rather acting downstream of them. However, heat can directly activate a cGMP pathway that opens the TAX-2/TAX-4 ion channel to increase calcium currents in neurons [59,60]. Future work should further help to elucidate additional molecular mechanisms.

## Methods

### *C. elegans* maintenance

Nematodes were grown on NGM plates seeded with *Escherichia coli* OP50 strain at 20°C unless otherwise stated, according to standard methods [61].

### C. elegans strains

*C. elegans* strains used and generated in this study are the following:

AGD1054 *uthIs366 [hsf-1^{neuro} line#1 (rab-3p:: hsf-1^{neuro}); myo-2p::tdtomato]*

AGD1289* *uthIs368 [hsf-1^{neuro} line#2 (rab-3p:: hsf-1^{neuro}); myo-2p::tdtomato]*

AX5968 *tax-4(p678); dbEx834[pops-1::tax-4cDNA::sl2mcherry; cGFP]–tax-4* rescue in ASG neuron

BX115 *lin-15B&lin-15A(n765) X; waEx16 [fat-6p::GFP + lin-15(+)]*

BX156 *fat-6(tm331) IV; fat-7(wa36) V*

BW1935 *unc-119(ed3) III; ctIs43 [dbl-1p::GFP + dbl-1p::GFP::NLS + unc-119(+)]; him-5(e1490) V*

CX2948 *tax-4 (p678) III*

HA1842 *rtIs30 [fat-7p::GFP] probably on I*

JN1715 *peIs1715 [str-1p::mCasp-1 + unc-122p::GFP]*, caspase expression in AWB neuron

LW2436 *jjIs2277 [pCXT51(5*RLR::pes-10p(deleted)::GFP) + LiuFD61(mec-7p::RFP)] I or IV*

MBA1069 *icbIs5[chil-27p::GFP, col-12p::mCherry-pest] IV; tax-4(ks11) III.; icbEx246[pFD18 (tax-4::tax-4::wrmScarlet::SL2::unc-54),pRJM163 (bus-1::GFP), BJ36]*

MOC141 *uthIs366 [hsf-1^{neuro} line#1 (rab-3p:: hsf-1^{neuro}); myo-2p::tdtomato], outcrossed 3X*

MOC151 *uthIs368 [hsf-1^{neuro} line#2 (rab-3p:: hsf-1^{neuro}); myo-2p::tomato]; rtIs30 [fat-7p:: GFP]*

MOC180 *uthEx663 [Ex-hsf-1^{neuro} (rab-3p:: hsf-1^{neuro}); myo-2p::tdtomato]*

MOC193 *uthEx663 [Ex-hsf-1^{neuro} (rab-3p:: hsf-1^{neuro}); myo-2p::tdtomato]; rtIs30 [fat-7p:: GFP]*

MOC201 *uthEx663 [Ex-hsf-1^{neuro} (rab-3p:: hsf-1^{neuro}); myo-2p::tdtomato]; rtIs30 [fat-7p:: GFP]; rrf-3(pk1426) II*

MOC227 *uthIs66 [hsf-1*<sup>neuro)</sup> *line # 1(rab-3p:: hsf-1*<sup>neuro)</sup>*; myo-2p::tomato]; rtIs30 [fat-7p:: GFP]*

MOC229 *uthIs368 [hsf-1*<sup>neuro</sup> *line#2 (rab-3p:: hsf-1*<sup>neuro</sup>*); myo-2p::tomato]; jjIs2277 [pCXT51 (5*RLR::pes-10p(deleted)::GFP) + LiuFD61(mec-7p::RFP)] I or IV*

MOC232 *uthEx663 [Ex-hsf-1*<sup>neuro</sup> *(rab-3p:: hsf-1*<sup>neuro</sup>*); myo-2p::tdtomato]; rtIs30 [fat-7p:: GFP]; tax-4 (p678) III*

MOC235 *uthEx663 [Ex-hsf-1*<sup>neuro</sup> *(rab-3p:: hsf-1*<sup>neuro</sup>*); myo-2p::tdtomato]; rtIs30 [fat-7p:: GFP]; ttx-3(ks5) X*

MOC252 *uthIs68 [hsf-1*<sup>neuro</sup> *line#2 (rab-3p:: hsf-1*<sup>neuro</sup>*); myo-2p::tomato]; tax-2(p671) I*

MOC253 *uthIs368 [hsf-1*<sup>neuro</sup> *line#2(rab-3p:: hsf-1*<sup>neuro</sup>*); myo-2p::tomato]; tax-2(p671) I; dbl-1 (nk3) V*

MOC254 *uthIs368 [line#2 (rab-3p:: hsf-1*<sup>neuro</sup>*); myo-2p::tomato]; dbl-1(nk3) V*

MOC279 *dbl-1(nk3) V; rtIs30 [fat-7p::GFP]*

MOC293 *uthIs368 [hsf-1*<sup>neuro</sup> *line#2 (rab-3p:: hsf-1*<sup>neuro</sup>*); myo-2p::tomato]; tax-2(p694) I*

MOC297 *uthEx663 [Ex-hsf-1*<sup>neuro</sup> *(rab-3p:: hsf-1*<sup>neuro</sup>*); myo-2p::tdtomato]; rtIs30 [fat-7p:: GFP]; ttx-1(p767) V*

MOC299 *uthIs368 [hsf-1*<sup>neuro</sup> *line#2 (rab-3p:: hsf-1*<sup>neuro</sup>*); myo-2p::tdtomato]; waEx16 [fat-6p::GFP + lin-15(+)]*

MOC312 *tax-2(p671)* outcrossed 3 times (parental strain PR671)

MOC315 *tax-4 (p678) III; rtIs30 [fat-7p::GFP]; uthEx663[rab-3p:: hsf-1*<sup>neuro;</sup> *myo-2p::tdtomato; bicEx34 [trx-1p::tax-4cDNA; myo-3p::mcherry], tax-4* rescue in ASJ, line #1.1@5

MOC316 *tax-4 (p678) III; rtIs30 [fat-7p::GFP]; uthEx663[rab-3p:: hsf-1*<sup>neuro;</sup> *myo-2p::tdtomato]; bicEx35 [trx-1p::tax-4cDNA; myo-3p::mcherry], tax-4* rescue in ASJ, line #1.1

MOC322 *tax-4 (p678) III; rtIs30 [fat-7p::GFP]; uthEx663[rab-3p::; myo-2p::tdtomato]; bicEx41 [ceh-36p::tax-4cDNA; myo-3p::mcherry], tax-4* rescue in AWC, line #2.3@5

MOC327 *tax-4 (p678) III; rtIs30 [fat-7p::GFP]; uthEx663[rab-3p:: hsf-1*<sup>neuro;</sup> *myo-2p::tdtomato]; bicEx46 [str-3p::tax-4cDNA; myo-3p::mcherry], tax-4* rescue in ASI, line #3.6@5

MOC328 *tax-4 (p678) III; rtIs30 [fat-7p::GFP]; uthEx663[rab-3p:: hsf-1*<sup>neuro;</sup> *myo-2p::tdtomato]; bicEx47 [trx-1p::tax-4cDNA; ceh-36p::tax-4cDNA; myo-3p::mcherry], tax-4* rescue in ASJ+AWC, line #4.4

MOC329 *tax-4 (p678) III; rtIs30 [fat-7p::GFP]; uthEx663[rab-3p:: hsf-1*<sup>neuro;</sup> *myo-2p::tdtomato]; bicEx48 [trx-1p::tax-4cDNA; ceh-36p::tax-4cDNA; myo-3p::mcherry], tax-4* rescue in ASJ+AWC, line #4.6

MOC331 *tax-4 (p678) III; rtIs30 [fat-7p::GFP]; uthEx663[rab-3p:: hsf-1*<sup>neuro;</sup> *myo-2p::tdtomato]; bicEx50 [trx-1p::tax-4cDNA; str-3p::tax-4cDNA; myo-3p::mcherry],* tax-4 rescue in ASJ +ASI, line #5.2

MOC332 *tax-4 (p678) III; rtIs30 [fat-7p::GFP]; uthEx663[rab-3p:: hsf-1*<sup>neuro;</sup> *myo-2p::tdtomato]; bicEx51 [ceh-36p::tax-4cDNA; str-3p::tax-4cDNA; myo-3p::mcherry], tax-4* rescue in AWC +ASI line #6.2

MOC333 *tax-4 (p678) III; rtIs30 [fat-7p::GFP]; uthEx663[rab-3p:: hsf-1*<sup>neuro;</sup> *myo-2p::tdtomato]; bicEx52 [trx-1p::tax-4cDNA; ceh-36p::tax-4cDNA; str-3p::tax-4cDNA; myo-3p::mcherry], tax-4* rescue in ASJ +AWC +ASI, line #7.4

MOC335 *tax-4 (p678) III; rtIs30 [fat-7p::GFP]; uthEx663[rab-3p:: hsf-1*<sup>neuro;</sup> *myo-2p::tdtomato]; bicEx54 [str-3p::tax-4cDNA; myo-3p::mcherry], tax-4* rescue in ASI, line #3.4@5

MOC340 *xuEx2070[Ptrx-1::TeTx::sl2::yfp; uthEx663[rab-3p:: hsf-1*<sup>neuro;</sup> *myo-2p::tdtomato],* tetanus toxin in ASJ neuron, blocks neurotransmission from ASJ

MOC342 *peIs1715 [str-1p::mCasp-1 + unc-122p::GFP, uthEx663[rab-3p:: hsf-1*<sup>neuro;</sup> *myo-2p::tdtomato],* AWB neuron eliminated

MOC343 *qrIs2 [sra-9::mCasp1] + intestinal GFP marker; uthEx663[rab-3p:: hsf-1*neuro; *myo-2p::tdtomato]*, ASK neuron eliminated

MOC346 *tax-4(p678) III; dbEx834[pops-1::tax-4cDNA::sl2mcherry; ccGFP]; rtIs30 [fat-7p:: GFP]; uthEx663[rab-3p:: hsf-1*neuro; *myo-2p::tdtomato]*, *tax-4* rescue in ASG

MOC347 *dbl-1(nk3) V; uthIs368[rab-3p:: hsf-1*neuro; *myo-2p::tomato]; rtIs30 [fat7p::GFP]*

MOC353 *rtIs30[fat-7p::GFP]; bicEx55 [oLC04 tax-4p::hsf-1(cDNA)::unc-54 3′ UTR; myo-2p::RFP] line 2*

MOC354 *rtIs30[fat-7p::GFP]; bicEx55 [oLC04 tax-4p::hsf-1(cDNA)::unc-54 3′ UTR; myo-2p::RFP] line 1*

MOC367 *bicEx68[tax-4p::his-24::morange (pAS1); bus-1p::GFP] syIs337 [15xUAS::?pes-10:: GFP::let-858 3′ UTR + ttx-3p::RFP + 1kb DNA ladder (NEB)] III. syIs400 [hsp16.41p::NLS:: GAL4SK::VP64::let-858 3′ UTR + unc-122p::RFP + 1kb DNA ladder (NEB)] V line 1*

MOC372 *uthIs368[rab-3p::hsf-1*neuro; *myo-2p::tomato]; ctIs43[dbl-1p:;GFP;dbl-1p::GFP:: NLS]*

MOC377 *Ex[dbl-1p::dbl-1::mcherry; unc-122p::GFP]; uthEx663[rab-3p::hsf-1; myo-2p::tdtomato]; rtIs30[fat-7p::GFP]*

MOC381 *icbEx246[pFD18(tax-4::tax-4::wrmScarlet::SL2::unc-54),pRJM163 (bus-1::GFP), BJ36]; syIs337 [15xUAS::?pes-10::GFP::let-858 3′ UTR + ttx-3p::RFP + 1kb DNA ladder(NEB)] III. syIs400 [hsp16.41p::NLS::GAL4SK::VP64::let-858 3′ UTR + unc-122p::RFP + 1kb DNA ladder(NEB)] V*

N2 *wild type*

NL2099 *rrf-3(pk1426) II*

NU3 *dbl-1(nk3) V*

PR671 *tax-2(p671) I*

PR694 *tax-2(p694) I*

PR767 *ttx-1(p767) V*

PS6025 *qrIs2 [sra-9::mCasp1] + intestinal GFP marker*, caspase expression in ASK neuron

PS7167 *syIs396 [unc-47p::NLS::NLS::GAL4SK::VP64::let-858 3′ UTR + unc-122p::RFP + 1kb DNA ladder (NEB)]; syIs337 [15xUAS::?pes-10::GFP::let-858 3′ UTR + ttx-3p::RFP + 1kb DNA ladder(NEB)] III*

PS7171 *syIs400 [hsp16.41p::NLS::GAL4SK::VP64::let-858 3′ UTR + unc-122p::RFP + 1kb DNA ladder(NEB)] V; syIs337 [15xUAS::?pes-10::GFP::let-858 3′ UTR + ttx-3p::RFP + 1kb DNA ladder(NEB)] III*

TQ6081 *xuEx2070[Ptrx-1::TeTx::sl2::yfp*, tetanus toxin in ASJ neuron—blocks neurotransmission from ASJ

TU3401 *sid-1(pk3321) V; uIs69 [pCFJ190(myo-2p::mCherry) + unc-119p::sid-1]* V–hypersensitive neuronal RNAi by feeding.

ZC273 *yxEx195[dbl-1p::dbl-1::TM::mcherry; unc-122p::gfp]*

QK52 *rde-1(n219); xkIs99 [wrt-2p::rde-1::unc-54 3′ UTR]*

* We noticed that the strain AGD1289 would sometimes be silenced, so the strain was rethawed on a monthly basis.

## Worm maintenance and synchronization

Unless stated otherwise, worms were grown at 20˚C. Worms were age synchronized either by egg-lay in a 2-hour period or by treatment with alkaline hypochlorite solution, according to standard procedures [62] for experiments that required large amounts of synchronized animals (such as the BODIPY staining, RNA sequencing [RNA-seq], and lipidomics experiments). WT N2 young adult animals were collected at 84 hours, 52 hours, and 45 hours post

L1 plating at 15˚C, 20˚C and 25˚C, respectively. The timing of mutant animals collected in parallel was adjusted from the WT timing, when developmental delay was noticed. For bulk quantitative RT-PCR (qRT-PCR) experiments and for microscopy-based quantification of fluorescent reporters at L4.8 or L4.9 stages, we used worms synchronized by egg laying and grown in parallel at different temperatures. Worms were harvested 97 hours after egg laying at 16˚C, 42 hours after egg laying at 25˚C, and L4.8 or L4.9 worms were picked from a mixed population grown at 20˚C. The L4 substages were assessed according to the morphology of the vulva, as described in [63]. In general, BODIPY and *fat-7p*::*GFP* fluorescence were monitored at young adult stage, except for *fat-7p*::*GFP* in *Ex hsf-1^neuro^* (**Fig 3A–3C** and **3H**), and *fat-6p*::*gfp* experiments, which was performed at day 3 of adulthood (**Fig 2J**).

## Experimental design

Each experiment was performed in at least 3 biological replicates. At least 30 individual animals were used in each replicate for microscopy experiments (BODIPY quantification, fluorescent reporter quantification), 15 to 20 animals in confocal experiments (RAD-SMAD reporter quantification), 15 to 30 animals in qRT-PCR experiments, 10 animals for pharyngeal assay, and 20 animals for fertility assay.

## Pharyngeal pumping assay

Synchronized young adult animals from each genotype were singled out and assayed for pharyngeal pumping at the young adult stage. Experiments were done in triplicate with at least 10 worms per condition. Pharyngeal pumping movements were followed for 30 seconds, under a dissection microscope, and each animal was scored at least twice.

## Fertility assay

About 20 worms per condition were singled out in 12-well plates seeded with 50-µL OP50 at the L4 stage. Each day, all animals were passed onto new 12-well plates. The F1 progeny laid by each individual worm was scored 2 to 3 days after the P0 parent had been transferred to the well, when the F1s were either L4 or adults. For ease of scoring, the 12-well plate was left on ice for a few minutes, until the animals were immobilized.

## Life span assay

Assays in **Fig 1** were performed at either 15˚C or 25˚C as previously described [24]. Worms were synchronized by egg laying within 2 hours. A total of 50 hermaphrodites were cultured on each 6-cm NGM petri dish, seeded with OP50. Animals were transferred to fresh plates every 1 to 2 days until the cessation of progeny production and every 2 to 3 thereafter, and scored every 2 to 3 days. Animals were scored as dead if they showed no spontaneous movement or response when probed on the nose. Animals dead from matricide (bagging), extruded intestine, desiccation on the side of the plate were censored. We noticed that *hsf-1^neuro^* animals tend to bag more often and also tend to leave the plate and desiccate; therefore, more data had to be censored. Statistical analysis was performed using GraphPad PRISM version 8. At least 3 biological replicates were obtained for each condition (**Table 1**). Life span assays in **S2 Fig** were conducted by SunyBiotech (Fuzhou, China) at 25˚C similar to the conditions described above except the number of live worms was assessed every day. In **S7 Fig**, life span experiments were performed on plates containing 25-µm FUDR (5-Fluoro-2'deoxyuridine thymidylate synthase inhibitor) as *dbl-1(nk3)* mutants are more susceptible to slightly pathogenic OP50 [64].

## DNA lysate preparation and PCR genotyping

Between 10 and 100 worms were picked into 10 μL of Worm Lysis Buffer (50mM KCl, 10mM Tris (pH 8.3), 2.5mM MgCl2, 0.45% NP40, 0.45% Tween-20, 0.01% Gelatin). Tubes were freeze-thawed once before 1 μL 01mg/mL of Proteinase K was added to each tube. Worms were lysed, and their genomic DNA was released by heating tubes to 65˚C for 60 to 90 minutes. Proteinase K was inactivated by heating to 95˚C for 15 minutes. Commonly, 1 μL DNA lysate was added to each PCR reaction. All PCR genotyping reactions were performed with Taq DNA polymerase (New England Biolabs (Ipswich, USA), #M0267L) with Thermopol buffer according to the manufacturer's instructions. The list of PCR primers used for genotyping can be found in **S12 Table**.

## RNAi assays

The RNAi suppressor screen (**Fig 3A**) was designed to look for an increase in the GFP fluorescent levels from *fat-7p*::*GFP* levels in animals carrying an overexpression of *hsf-1^neuro^* in neurons. As neurons tend to be refractive to RNAi, the screen was performed in the MOC201 strain carrying a Lof allele in *rrf-3(pk1426)*, which enhances RNAi sensitivity, including in neurons [65]. Because *fat-7* expression levels are highly sensitive to dietary variations, the screen was performed using an extrachromosomal array, which carried the transgene *rab-3p*::*hsf-1^neuro^*. In this experimental setup, due to the incomplete transmission of extrachromosomal arrays, siblings of different genotypes were grown side by side under identical conditions. The effect of RNAi treatment on *fat-7p*::*GFP* in siblings that inherited the array was compared to those that had not. RNAi-mediated KD of candidate genes involved in neuronal functioning was performed using clones from Dr Julie Ahringer's RNAi library, including *tax-4*, *ttx-3*, *cat-2*, *tax-6*, *tbh-1*, *tph-1*, and *unc-31*. The similarity to human genes and some of the known functions of these genes were taken from Wormbase and listed below:

- *tax-4*/ **F36F2.5** is an ortholog of human cyclic nucleotide gated channel subunit beta 3 and contributes to cGMP-activated cation channel activity in sensory neurons.

- *ttx-3*/**C40H5.5** is an ortholog of human LIM homeobox 9. It is involved in several processes, including generation of neurons and thermotaxis.

- *cat-2*/**B0432.5** is an ortholog of human tyrosine hydroxylase. It is involved in several processes, including dopamine biosynthetic process from tyrosine, habituation, and turning behavior involved in mating.

- *tax-6*/**C02F4.2** is an ortholog of human protein phosphatase 3 catalytic subunit. It has calcium–ion binding activity and calmodulin-dependent protein phosphatase activity. It is involved in male mating behavior and growth.

- *tbh-1*/**H13N06.6** is an ortholog of human dopamine beta-hydroxylase. It is involved in octopamine biosynthesis and localizes to RICL and RICR neurons and gonad.

- *tph-1*/ **ZK1290.2** is an ortholog of human tryptophan hydroxylase 1. It is involved in several processes, including synthesis of serotonin biosynthetic from tryptophan.

- *unc-31*/ **ZK897.1** is an ortholog of human calcium–dependent secretion activator. It is involved in several processes, including nervous system development and secretion of neurotransmitters.

Bacterial cultures were grown overnight in LB with 100 μg/mL Carbenicillin and induced with 1 mM IPTG for 2 hours. RNAi seeded plates were left for 48 hours to dry at room

temperature. Animals were initially grown on OP50 plates and were individually transferred at the L4.8 stage [63] onto RNAi plates. About 50 extrachromosomal overexpressing neuronal *hsf-1^neuro* worms and non-extrachromosomal worms from the same background were transferred onto the same RNAi plate. Animals were transferred onto fresh RNAi plates at day 2 of adulthood. At day 3 of adulthood, the animals were mounted, and *fat-7p*::GFP fluorescence was monitored. We tried to image about 40 extrachromosomal carrying neuronal *hsf-1^neuro* overexpression array and 40 non-extrachromosomal siblings. At least 3 biological replicates of each experiment were performed. For *hsf-1^neuro*RNAi exclusively in neurons (**Fig 6F–6I**), TU3401 animals, hypersensitive to RNAi in neurons [66], were grown on OP50 plates from eggs to late-L4 stage. L4.8 stage animals were transferred onto L4440 (EV) or RNAi plates and harvested for qRT-PCR 48 hours later at day3 of adulthood. Some leakage of the *unc-119* promoter has been observed in the hypodermis [67], and to control for this caveat, we tested the effect of *hsf-1* RNAi in the hypodermis. *rde-1(219)* mutant animals carrying the *rde-1* rescue gene under hypodermis specific *wrt-2* promoter (QK52) were grown and harvested similar as for TU3401 animals. To control for the effectiveness of the RNAi treatment, animals were subject to heat shock, and the expression of HSPs was monitored. All results are listed in **S8 Table**. The dsRNA expressing plasmid containing *hsf-1*came from Dr J. Ahringer RNAi library.

## qRT-PCR on bulk worm samples

To monitor steady-state mRNA levels on bulk samples of worms, we used the Power SYBR Green Cells-to-$C_T$ kit and hand-picked a pool of about 15 to 25 animals in 10 μL of Lysis buffer. Reverse transcription was performed using the 2X RT buffer from the Power SYBR Green Cells-to-$C_T$ kit according to the manufacturer's instructions, and cDNA was diluted either 1:4 or 1:5. Each qRT-PCR reaction contained 1.5 μL of primer mix forward and reverse at 1.6 μM each, 3.5 μL of nuclease free water, 6 μL of 2X Platinum SYBR Green qPCR Supermix-UDG with ROX (ref 11744–500), and 1 μL of diluted cDNA. The list of PCR primers used for qRT-PCR is in **S10 Table**. The PCR efficiency was calculated for each couple of primers by running a standard curve on a dilution series. Validated couples of primers had a PCR efficiency between 90 and 113% with $R^2 > 0.98$ (**S12 Table**). Expression levels of steady-state mRNA were calculated using the ΔCt method. Each target mRNA was normalized to the average of the 3 housekeeping genes: *cdc-42*, *ire-1*, and *pmp-3*. Measurements of mRNA levels shown in **S1A Fig** and **Figs 3J and 5E** were obtained by qRT-PCR on a regular thermocycler (CFX96 or CFX384 real time system, Bio-Rad, Hercules, USA), whereas mRNA levels measured in **Fig 3D and 3E** and **Fig 5F–5I** were obtained on a high-throughput thermocycler using nano-fluidic chips (Biomark HD, Fluidigm, San Francisco, USA), according to the protocol we have optimized for bulk qRT-PCR [68].

## Generation of transgenic lines

In order to dissect the involvement of ASJ, ASI, and AWC neurons in *hsf-1^neuro*–dependent fat remodeling across tissues, we generated extrachromosomal transgenic lines carrying the following plasmids kindly provided by Kuhara and colleagues [59]: pTOM003 [*str-3p*::*tax-4cDNA*] (*tax-4* rescue in **ASI** neuron), pTOM004 [*trx-1p*::*tax-4cDNA*] (*tax-4* rescue in **ASJ** neuron), and pTOM010 [*ceh-36p*::*tax-4cDNA*] (*tax-4* rescue in **AWC** neuron). Injections were performed by Invermis at Imperial College London. The plasmids were injected in MOC232 *uthEx663 [Ex- hsf-1^neuro (rab-3p:: hsf-1^neuro); myo-2p::tdtomato]; rtIs30 [fat-7p::GFP]; tax-4 (p678)*, at a final concentration of 10 ng/μL together with *myo-3p::mcherry* co-injection marker at 5 ng/μL and random DNA up to 100 ng/μL (from linearized empty backbone BJ36). When 2 or 3 plasmids were co-injected for *tax-4* rescued expression in a combination of neurons, the

plasmids were injected at a lower final concentration of 5 ng/μL each plasmid to avoid potential toxicity issues. For *tax-4* expressing neurons co-localization experiments (**S4B Fig**), the plasmid pAS1[*tax-4p::his-24::mOrange*], a gift from Andrew Leifer (Addgene plasmid #124340) was injected at 10 ng/μL together with the co-injection marker *bus-1*p::GFP at 30 ng/μL by Magnitude Biosciences at Durham University in PS7171 animals. To generate extrachromosomal lines overexpressing *hsf-1* in *tax-4* neurons only (**S5 Fig**), the construct [*tax-4p::hsf-1(cDNA)*] was generated by SunyBiotech by cloning 3-kb upstream the TSS of *tax-4* driving *hsf-1* CDNA, to which an artificial intro was added to optimize expression. Both fragments were cloned in pPD49.78 vector carrying *unc-54* 3′ UTR (760 bp). The *tax-4* promoter is flanked by enzymatic restriction sites HindIII in 5′ and EcoRI site in 3′, while *hsf-1* cDNA is flanked by EcoRI and EcoRV restriction sites in 5′ and 3′, respectively. The [*tax-4p::hsf-1(cDNA)*] construct obtained was then injected by Magnitude Biosciences (Durham, UK) at 10ng/μL together with *myo-2*p::RFP co-injection marker at 5 ng/μL in HA1842 animals.

## Worm harvesting for lipidomics

Worms of every genotype were synchronized using hypochlorite treatment according to standard procedures [62]. Young adult animals were grown at 20˚C and harvested at 50 to 52 hours post L1 plating for N2, MOC141, and NU3, at 54 hours for AGD1289, and at 70 hours for BX156, as they were developmentally delayed. One 9-cm NGM plate containing either 1,000 young adults for total FA lipidomics or 500 worms for phospholipids analysis was harvested for each genotype. Worms were washed with 15 mL of M9 buffer at least 3 times. Most of the supernatant was removed, and the pellets of collected worms were frozen at −80˚C in low protein binding Eppendorf tubes before being processed for lipidomics analysis.

## GC–MS analysis of FAs

A total of 16 to 35 mg of *C. elegans* were extracted by adding 1 mL chloroform/methanol (2:1, v: v, containing 0.01% BHT and 10 μg of C9:0 and C13:0, respectively, as internal standard) and processed using an ultrasound sonotrode for 30 seconds at 40 Hz (type UW 2070, Bandelin, Berlin, Germany). Afterward, 0.5 mL water was added to each tube, and each sample was shaken vigorously for 1 minute. Next, the extract was centrifuged for 5 minutes at 3,000 rpm. The chloroform layer was transferred into a new vial, and the solvent was removed with a gentle stream of nitrogen. The residue was resolved in 200 μl tetrahydrofuran. Moreover, 400-μl methanolic base 0.5 M (Acros Organics, Geel, Belgium) was added. After 1 minute of shaking, the sample was heated for 15 minutes at 80˚C. Afterward, 200 μL water and 200 μl hexane were added. After another minute of vigorous shaking, the sample was centrifuged for 1 minute at 1,500 rpm. The hexane layer was transferred into a new vial. Furthermore, 1 μL sample was injected into the GC–MS (TQ8040, Shimadzu, Kyoto, Japan) with a split of 5 and injection temperature of 260˚C. The separation was performed with a Zebron ZB-5MSplus column (30 m × 0.25 mm × 0.25 μm) (Phenomenex, Torrance, USA) with helium as carrier gas: 35˚C was held for 2 minutes. Then the temperature was raised by 10˚C per minute to 140˚C, which was held for 10 minutes. Afterward, the temperature was raised to 240˚C at a rate of 2˚C per minute; 240˚C was held for 10.5 minutes. Quantification was performed using the GCMSSolution Version 4.30 (Shimadzu). The lipidomics data were quantified and analyzed as described in [69].

## Phospholipids analysis

The worms were homogenized using a Precellys evolution with a cryolys unit to keep the sample frozen during homogenization (Bertin Technologies, France). Precellys bead-beating tubes

with reinforced walls for hard tissue (CRK28-R) were used with 3 cycles of 7,200 rpm at 45 seconds. The homogenized sample was then transferred to a glass tube containing Chloroform/Methanol for lipid extraction using Folch method [70]. Phospholipids were separated using a Cogent HPLC column (150 × 2.1 mm, 4 μm particle size) placed on a Shimadzu XR (Shimadzu) using the conditions described in [71]. The phospholipids were then detected using an Orbitrap Elite mass spectrometer in full scan mode with a mass range of 200 to 1,000 *m/z* at a target resolution of 240,000 (FWHM at *m/z* 400). Data were analyzed using Lipid Data Analyzer (2.6.0–2) software [72]. The desaturation index was calculated for each phospholipid species (e.g., PI38, PI with an acyl chain of 38 carbons) as the sum of the weight percentage of each subspecies (e.g., PI38:3, PI38 with 3 double bonds) multiplied by the number of double bonds of this particular subspecies (3 in the case of PI38:3), similar to [17].

## Fat content analysis using BODIPY

We used BODIPY 493/503 (Thermo Fisher Scientific (Waltham, USA), D3922) to stain neutral lipids in fixed animals. We adapted a published protocol [73] to fix worms using 60% isopropanol. Our protocol is described in [53]. Briefly, about 1,000 worms per genotype were synchronized by hypochlorite treatment [62]. Animals were collected and washed at least 3 times in M9 buffer. Worms were then fixed for 5 minutes in 60% isopropanol in 1.5-mL protein low-bind Eppendorf tubes, with occasional inversion of the tubes. As fixation with isopropanol was sometimes variable, we also tried fixation with cold methanol for 10 minutes, the rest of the protocol remaining identical. After fixation, we let the worms settle by gravity and washed them once more with M9. The M9 supernatant was removed, leaving approximately 50 μL. The tubes were frozen and thawed twice and then incubated with 500 μL of BODIPY 493/503 (diluted in M9 at 1 μg/mL) at room temperature for 1 hour on a rotator. After 1 hour, the worms were washed twice with 1 mL M9 solution containing 0.01% triton. We kept the samples at 4˚C and imaged them either the same day or the next day but not more than 2 to 3 days after collection.

## Microscopy

All worms imaged were mounted on a 2% agarose pad using a 18 × 18 mm glass coverslip, protocol detailed in [53]. For imaging of live worms, animals were paralyzed in 3mM Levamisole diluted in M9. Before mounting fixed animals (BODIPY protocol), we used a mouth pipette with a glass capillary to remove all liquid in the Eppendorf tube. About 8 to 10 μL of Vectashield antifade mounting media without DAPI (Vector Laboratories (Burlingame, USA), 94010) was added. Fluorescence exposure was identical across all conditions of the same experiment. To image intestinal levels of *fat-7p*::*GFP* fluorescence in live worms and BODIPY 493/503 fluorescence in fixed worms, we used a Nikon (Tokyo, Japan) Ti Eclipse fluorescent microscope at objective 20×. For imaging GFP-positive neurons in PS7171 and PS7167, we used a Nikon fluorescent stereomicroscope SMZ18, as it was easier to capture neurons in 2D from live animals, except for images shown in **Fig 6A and 6B**, which were obtained on a Nikon Ti Eclipse fluorescent microscope at objective 20×. PS7171 and PS7167 were synchronized at L4.8 stage [63] in this experiment. LW2436 and MOC229 worms were imaged on a Nikon A1R confocal microscope at 20× objective to determine the fluorescence levels of the RAD-SMAD reporter. The Z-stacks taken were then stitched together. The stage at which animals were imaged is detailed in the legends of each figure. Animals carrying either the *tax-4p*::*tax-4*::*wrmScarlet* or *tax-4p*::*his-24*::*mOrange* transgenes were imaged on a Zeiss confocal microscope at 60× objective.

## RNA-seq library preparation

RNA was extracted by standard Trizol extraction techniques. Libraries were made using either NEBNext mRNA Second Strand Synthesis Module (E6111) followed by NEBNext Ultra II DNA Library Prep Kit for Illumina (NEB-E7645) or NEBNext Ultra II Directional RNA Library Prep Kit for Illumina (E7760) with the NEBNext Poly(A) mRNA Magnetic Isolation Module (NE7490) as per manufacturer's protocols using half reactions. A total of 13 cycles of amplification was used for library enrichment; quality and size distribution of the libraries were ascertained by running on a Bioanalyzer High Sensitivity DNA Chip (Agilent (Santa Clara, USA), 5067–4626), and concentration was determined using KAPA Library Quantification Kit (KK4824). Libraries were sequenced on an Illumina HiSeq 2500 system by the Babraham Sequencing Facility. The experiment compared 3 independent replicates WT (N2) and *hsf-1^{neuro}* (AGD1289) animals.

## Quantification and statistical analysis

**Statistical analysis.** Statistical information for each experiment can be found in the corresponding figure legend and was obtained with Prism 8.0.

**Fluorescence quantification and worm area measurements.** Fluorescence images of *fat-7p*::GFP reporter at young adult stage (**Figs 2H and 4D**) or day 3 adults (**Fig 3J**) and BODIPY fat staining images, all taken at young adult stage (**Figs 2E, 2G, 3F, and 4G**), were quantified using an in-house–designed semiautomated workflow [53]. In order to straighten and quantify fluorescence from acquired *C. elegans* images, we have developed a FIJI/ImageJ workflow called "Worm-align" that allows to generate single- or multichannel montage images of aligned worms from selected animals in the raw image. The output of "Worm-align" was then imported and run through a CellProfiler pipeline called "Worm_CP" for fluorescence quantification of the animals of interest selected with "Worm-align." One output parameter of the "Worm_CP" pipeline is the area measurement for every worm, which was used as a proxy of the size of the animals (**S2A Fig**). Both "Worm-align" and "Worm_CP" pipelines are available and described in [53]. For images of *fat-7p*::GFP reporter taken in animals at day 3 of adulthood (**Fig 3C and 3H**), and of *fat-6p*::GFP fluorescence at day 3 of adulthood (**Fig 2J**), *fat-7*p::GFP and *fat-6p*::GFP fluorescence was quantified manually in FIJI/ImageJ by circling every worm and a dark background zone in each image. The fluorescence intensity was measured in the region of interest (ROI) manager, and the value measured for the image background was subtracted from each individual worm fluorescence measurement, as described in [74]. GFP fluorescence from Z-stack confocal images of LW2436 and MOC229 animals, carrying the nuclear RAD-SMAD reporter (**Fig 4B and 4C**), was quantified using Fiji software. The RAD-SMAD reporter is expressed mainly in epidermal nuclei and in large intestinal nuclei. To discriminate between both tissues, GFP fluorescence was quantified either for large nuclei (>60 μm), corresponding to intestinal nuclei, or smaller nuclei (<60 μm), corresponding to epidermal nuclei. For imaging of the ctIs43 [dbl1-pGFP + dbl-1p::NLS] transgene (**S6B and S6D Fig**), a minimum of 10 Z-stacks were acquired of individual worms on a Nikon Ti Eclipse inverted epifluorescence microscope (objective 20×) with a 0.8-μm step size across a 17.35-μm z-range. Stacks were processed and analyzed in FIJI [75]. In brief, image stacks were processed by removing any bright fluorescent head region (corresponding to the myo-2p::tdtomato co-injection marker of *hsf-1^{neuro}* animals) and applying a 3D White top-hat filter using the MorphoLibJ plugin [76]. Three-dimensional nuclei in the processed stacks were segmented and analyzed using the 3D objects counter [77].

**RNA-seq analysis.** The FASTQ files were quality trimmed with Trim Galore v0.4.4 (https://github.com/FelixKrueger/TrimGalore), used in conjunction with Cutadapt v1.15

(DOI: 10.14806/ej.17.1.200), and then mapped to the *C. elegans* reference genome WBcel235 using HISAT2 v2.1.0 [78] in either single- or paired-end mode depending on the sequencing protocol followed. To improve the mapping efficiency across splice junctions, HISAT2 took as additional input the list of WBcel235.75 known splice sites. RNA-seq QC and analysis was performed on the mapped reads using the genome browser SeqMonk v1.45.4 (https://www.bioinformatics.babraham.ac.uk/projects/seqmonk/). To identify DE genes, the number of reads positioned over exons were first calculated for every gene using the program's "RNA-seq Quantitation Pipeline." DE genes were subsequently called by DESeq2 v1.22.2 [79], launched from SeqMonk with default settings. DE genes were defined by having an adjusted *p*-value cutoff <0.05 after multiple testing correction. The gene expression principal component analysis when comparing *hsf-1^neuro^* and WT showed that the transcriptomes formed clusters according to their molecular subtypes, indicating high-quality and consistent homogeneity of transcriptomes.

**Gene overlap.** In order to determine if there was an enrichment between DE-seq genes in WT at 25˚C versus WT at 15˚C, and *hsf-1^neuro^* versus WT (at 20˚C), raw data from [29] were reanalyzed using DESeq2 with a $p < 0.05$ cutoff, which gave a total of 1,089 genes, and genes DE between WT and *hsf-1^neuro^* #2 (DESeq2 using a cutoff $p < 0.005$, which gave 2,136 genes. A hypergeometric distribution was employed to determine the probability of overlap when using different filtering criteria (0, 0.05, 1, 1.5, or 2 log2-fold change).

## Supporting information

**S1 Fig. Overexpression of *hsf-1* in neurons does not cause animals to experience starvation.** HSF-1, heat shock factor 1.
(DOCX)

**S2 Fig. Lof mutation in *tax-2* suppresses life span extension of *hsf-1^neuro^* at 25˚C.** HSF-1, heat shock factor 1; *hsf-1^neuro^*, neuronal overexpression of *hsf-1*; Lof, loss of function.
(DOCX)

**S3 Fig. Overexpression of *hsf-1* in *tax-4* expressing neurons is able to rescue *fat-7p::GFP* fluorescence but not lipases levels.** HSF-1, heat shock factor 1.
(DOCX)

**S4 Fig. Overexpression of *hsf-1* in neurons phenocopies TGF-β/BMP mutants.** BMP, bone morphogenetic protein; HSF-1, heat shock factor 1; TGF-β, transforming growth factor ß.
(DOCX)

**S5 Fig. *hsf-1^neuro^* decreases the activity of TGF-β/BMP signaling.** BMP, bone morphogenetic protein; HSF-1, heat shock factor 1; *hsf-1^neuro^*, neuronal overexpression of *hsf-1*; TGF-β, transforming growth factor ß.
(DOCX)

**S6 Fig. (A)** Neurons expressing *tax-4p::tax-4::wrmScarlet* are adjacent to neurons where HSF-1 is activated. **(B)** Neurons expressing *tax-4p::his-24::mOrange* are adjacent to neurons where HSF-1 is activated, with occasional overlap. **(C)** Epistasis analysis suggest that *tax-2* plays roles beyond the modulation of TGF-β signaling. HSF-1, heat shock factor 1; TGF-β, transforming growth factor ß.
(DOCX)

**S7 Fig. The activity of TGF-β/BMP and fat desaturases is detrimental to survival at 25˚C.** BMP, bone morphogenetic protein; TGF-β, transforming growth factor ß.
(DOCX)

**S8 Fig. Neuronal and not epidermis expression of *hsf-1* is necessary for lipase expression at 25˚C.** HSF-1, heat shock factor 1.
(DOCX)

**S9 Fig. An HSF-1–dependent thermostat to control organismal adaptation to warmer temperatures.** HSF-1, heat shock factor 1.
(DOCX)

**S1 Table. PLs lipidomics analysis.** PL, glycerophospholipid.
(XLSX)

**S2 Table. Desaturation index calculated for PLs lipidomics analysis.** PL, glycerophospholipid.
(XLSX)

**S3 Table. DE genes from RNA-seq *hsf-1^{neuro}* versus wt.** DE, differentially expressed; HSF-1, heat shock factor 1; *hsf-1^{neuro}*, neuronal overexpression of *hsf-1*; RNA-seq, RNA sequencing; WT, wild-type.
(XLSX)

**S4 Table. Probability of overlap calculated using a hypergeometric distribution between DE genes from AGD1289 (*hsf-1^{neuro}*#2) versus N2 and N2 at 25˚C versus N2 at 15˚C.** DE, differentially expressed; HSF-1, heat shock factor 1; *hsf-1^{neuro}*, neuronal overexpression of *hsf-1*.
(DOCX)

**S5 Table. BODIPY intensity measurements.**
(DOCX)

**S6 Table. Quantification of fluorescent reporters.**
(XLSX)

**S7 Table. Free FAs lipidomics analysis.** FA, fatty acid.
(XLSX)

**S8 Table. Gene expression analysis.**
(XLSX)

**S9 Table. The activation of *hsf-1^{neuro}* in 6 or more *tax-2/tax-4* expressing neurons is required for remote fat remodeling.** HSF-1, heat shock factor 1; *hsf-1^{neuro}*, neuronal overexpression of *hsf-1*.
(DOCX)

**S10 Table. Fertility assays.**
(DOCX)

**S11 Table. Number of GFP-positive neurons in PS7171 (*hsp16.41p*::cGAL/UASp::GFP) at different growth temperatures.**
(DOCX)

**S12 Table. List of primers used in this study.**
(DOCX)

**S13 Table. Life span assays data.**
(XLSX)

## Acknowledgments

We dedicate this work to the memory of Michael J.O. Wakelam.

We would like to acknowledge Michael Fasseas (Invermis, Magnitude Biosciences) for plasmid injections and Sunny Biotech for transgenics; Catalina Vallejos and John Marioni for statistical advice at the beginning of the work; Simon Walker, Imaging, Bioinformatics and Lipidomics Facilities at Babraham Institute for technical support; and Cindy Voisine, Michael Witting, Jon Houseley, Len Stephens, Carmen Nussbaum Krammer, Rebeca Aldunate, Patricija van Oosten-Hawle, Jean-Louis Bessereau, and Jane Alfred for feedback on the manuscript. We thank Andy Dillin, Atsushi Kuhara, Amy Walker, Andrew Leifer, Yun Zhang, and Michalis Barkoulas for reagents and Julie Ahringer, Anne Ferguson-Smith, and Anne Corcoran for support and helpful discussions. We also acknowledge Babraham Institute Facilities.

## Contact for reagent and resource sharing

Further information and requests for resources and reagents should be directed to and will be fulfilled by the Lead Contact, Olivia Casanueva (moc771@gmail.com).

## Author Contributions

**Conceptualization:** Laetitia Chauve, Sharlene Murdoch, Michael J. O. Wakelam, Olivia Casanueva.

**Data curation:** Laetitia Chauve, Francesca Hodge, Sharlene Murdoch, Andrea F. Lopez-Clavijo, Olivia Casanueva.

**Formal analysis:** Laetitia Chauve, Anne Segonds-Pichon, Steven W. Wingett, Olivia Casanueva.

**Funding acquisition:** Olivia Casanueva.

**Investigation:** Laetitia Chauve, Francesca Hodge, Sharlene Murdoch, Fatemeh Masoudzadeh, Harry-Jack Mann, Cheryl Li, Olivia Casanueva.

**Methodology:** Laetitia Chauve, Francesca Hodge, Sharlene Murdoch, Andrea F. Lopez-Clavijo, Hanneke Okkenhaug, Greg West, Bebiana C. Sousa, Hermine Kienberger, Karin Kleigrewe, Mario de Bono.

**Project administration:** Francesca Hodge, Sharlene Murdoch, Olivia Casanueva.

**Resources:** Laetitia Chauve, Olivia Casanueva.

**Supervision:** Laetitia Chauve, Sharlene Murdoch, Olivia Casanueva.

**Validation:** Olivia Casanueva.

**Visualization:** Laetitia Chauve, Francesca Hodge, Sharlene Murdoch, Harry-Jack Mann, Olivia Casanueva.

**Writing – original draft:** Olivia Casanueva.

**Writing – review & editing:** Laetitia Chauve, Olivia Casanueva.

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
