## [Editor Report · Decision Letter 0]

11 Jan 2021

Dear Dr Casanueva, 

Thank you for submitting your manuscript entitled "Neuronal HSF-1 activity coordinates fat desaturation across tissues in Caenorhabditis elegans via TGF-β/BMP signalling" for consideration as a Research Article by PLOS Biology. 

Your manuscript has now been evaluated by the PLOS Biology editorial staff as well as by an academic editor with relevant expertise and I am writing to let you know that we would like to send your submission out for external peer review.

Please re-submit your manuscript within two working days, i.e. by Jan 13 2021 11:59PM.

We also do apologize for the time it has taken to send you our initial decision - we were out of the office over the Holidays and unfortunately we weren't quite able to finalize our discussion on your manuscript last week. Feel free to email us at plosbiology@plos.org if you have any queries relating to your submission.

Kind regards,

Lucas Smith, Ph.D.,

Associate Editor

PLOS Biology

---

## [Decision Letter · Decision Letter 1]

4 Mar 2021

Dear Dr Casanueva,

Thank you very much for submitting your manuscript "Neuronal HSF-1 activity coordinates fat desaturation across tissues in Caenorhabditis elegans via TGF-β/BMP signalling" for consideration as a Research Article at PLOS Biology. Your manuscript has been evaluated by the PLOS Biology editors, an Academic Editor with relevant expertise, and by several independent reviewers.

The reviews are appended below. As you will see, the reviewers find your results interesting, however they also raise a number of concerns that would need to be addressed, including that the main conclusions of the study are not adequately supported by the experiments performed at this stage. The reviewers suggest several new experiments and analyses that would be important to strengthen the study and provide more mechanistic insights. Having discussed the reviewers' comments with the Academic Editor, we think that the revised manuscript should, for example, provide insights into the role of endogenous HSF-1 in neurons (you could knockout hsf-1 in neurons to show it abrogates the effects) and provide stronger mechanistic insights to strengthen the link between hsf1 and BMP signaling. While we think that Reviewer 1’s point 4 regarding “cold sensors” is interesting, we do not think that inclusion of new data to address this point would be necessary for publication in PLOS Biology. If it is helpful, we would be happy to assess a revision plan for additional input.

In light of the reviews, we will not be able to accept the current version of the manuscript, but we would welcome re-submission of a much-revised version that takes into account the reviewers' comments. We cannot make any decision about publication until we have seen the revised manuscript and your response to the reviewers' comments. Your revised manuscript is also likely to be sent for further evaluation by the reviewers.

We expect to receive your revised manuscript within 3 months. 

**IMPORTANT - SUBMITTING YOUR REVISION**

*Re-submission Checklist*

*Published Peer Review*

*PLOS Data Policy*

*Blot and Gel Data Policy*

Sincerely,

Lucas Smith, Ph.D.,

Associate Editor,

lsmith@plos.org,

PLOS Biology

REVIEWS:

Reviewer #1: Using a C. elegans strain over-expressing hsf-1 in neurons, the authors of the manuscript comprehensively characterized phenotypic consequences of fat remodeling in non-neuronal peripheral tissues. Based mainly on these findings and activation of HSF-1 target gene reporters in neurons by chronic warmer temperature, the authors conclude that HSF-1 in neurons serve as a thermostat-based mechanism to cell non-autonomously coordinate membrane saturation and composition across tissues in a multicellular animal. While the observations are interesting, there are major issues of this paper that need to be addressed before suitable publication. 

1, Almost the entire paper is based on studies of the C. elegans strain over-expressing hsf-1 in neurons. How do conclusions from such studies say anything about the physiological roles of endogenous HSF-1 in neurons? Does loss of function of hsf-1 in neurons (by either Cre-LoxP deletion or neuronal rescued of hsf-1 KO, both are technically feasible) have any phenotypes in fat remodeling of peripheral tissues? Activation of HSF-1 target gene reporters in neurons by chronic warmer temperature is evidence of correlation in nature, providing no causal relationship between neuronal HSF-1 and fat remodeling in peripheral tissues.

2, Technically, how do authors exclude the possibilities that the so-called neuronal rab-3 promoter does not have residual activity in non-neuronal tissues responsible for the phenotypes observed? This is particularly a concern when high-copy number instead of single-copy transgene was used in this case, and the conclusion is based on a single "neuronal specific" promoter in an overexpression system. 

3, The authors used RAD-SMAD reporters to monitor regulation of BMP by HSF. This should be complemented with studies (QPCR and low-copy transgene etc) on the regulation of specific and endogenous BMP pathway genes (e.g. dbl-1) to convincingly support the model proposed. More importantly, it remains unclear whether such potential regulation of BMP/dbl-1 by HSF-1 actually contributes to fat remodeling. Can transgenic rab-3p::dbl-1 in nhsf-1 worms restore fat-7 expression and lipid saturation profiles?

4, Recent studies have demonstrated importance of several "cold sensors" in C. elegans involved in neuronal sensing of cold temperature and peripheral tissue lipid homeostasis. It would be important to test if those pathways (trpa-1 and paqr-2) act independently with HSF-1 or not in lipid remodeling.

Reviewer #2: In this important manuscript, Chauve and colleagues report the finding of a thermostat that cell non-autonomously coordinates global changes in membrane composition and metabolism allowing an ectotherm to adapt to higher temperatures and enhance survival in these conditions. 

The authors show that C. elegans worms that express hsf-1 panneuronally display transcriptomic and metabolic profiles similar to those displayed by animals grown at 25 ºC despite having grown at 15 ºC. This is a shift in fat desaturases and membrane lipid composition that requires the function of cGMP gated-channels tax-2 and tax-4, which are expressed in sensory neurons. Then, they go on to identify a group of six sensory neurons where tax-2 function is sufficient to drive the nhfs-1-associated changes in lipid composition. Finally, they identify the suppression of TGF beta signalling as one of the mechanisms by which the effects of nhfs-1 are implemented. 

In my opinion, the finding reported here is extremely novel and it changes the way we thought about how ectotherms regulate temperature-associated adaptive changes. The data has been carefully acquired, supports the conclusions stated and allows the authors to generate a model that, even if it may be incomplete for the time being since this is a novel finding, is informative and useful.

I only have one comment to make regarding the data of tax-4 presented in page 20 and figure 3C. The authors mention that disruption of tax-4 (in p678) "does not significantly alter the output of fat-7::gfp reporter in wildtype animals" but there is no statistical analysis for this comparison. They also say that the effect of nhsf-1 on lipid enzymes is dependent on tax-4 and that both genes act in a linear pathway. However, nhsf-1 can still significantly reduce fat-7::gfp levels in tax-4 mutants (figure C) and so, the interpretation is that nhsf-1 acts through tax-4 dependent and tax-4 independent pathways. 

Reviewer #3: 

The authors present impressive amount of data to show that neuronal HSF-1 signaling coordinately remodels lipid composition across tissues to adapt to thermal variation. The authors clearly show that overexpression HSF-1 in neurons induces lipid desaturation in a cell non-autonomous manner. The effect of neuronal HSF-1 expression on fat desaturation requires (at least in part) the cyclic nucleotide gated cation channel TAX-2/TAX-4 in sensory neurons results in downregulation of TGF-β/BMP signaling to modulate fat metabolism. This work suggests a novel mechanism whereby lipid composition can be centrally regulated in a multicellular organism to facilitate membrane stability over a wide range of temperatures. This is an interesting and novel finding. However, I have some concerns to what extent the considerable amount of data support their model. 

Major points:

1. The authors present a model in which hsf-1 acts in the TAX-2/TAX-4 sensory neurons as a thermostat to control lipid remodeling and heat adaptation. However it is not clear whether hfs-1 expression in TAX-2/TAX-4 sensory neurons is sufficient for lipid remodeling and an increase lifespan at 25C. They should strengthen the argument by testing if the expression of hsf-1 specifically in those neurons (e.g. tax-2::hsf-1) is sufficient to control fat metabolism. The argument also should be strengthened by checking if HSR is activated in those six neurons at 25°C (e.g. hsp-16.41::GFP expression, hsf-1 expression, or hsf-1 nuclear localization).

2. The authors claim that overexpression of hsf-1 in neurons suppresses TGF-β/BMP signaling to remodel fat metabolism. Currently the link between HSF-1 and TGF-β/BMP signaling is tentative. The function of TGF-β/BMP signaling in regulating lipid metabolism including fat-6/fat-7 expression is well known (Luo et al., 2010; Yu et al., 2017; Clark et al., 2018). What remains unknown is how HSF-1 functions to reduce TGF-β/BMP signaling. dbl-1 is expressed mainly in neurons. It would be interesting to test if expression of dbl-1 in neurons (also in tax-2/tax-4 neurons?) is decreased at 25ºC and in the nhsf-1 animals even at 20ºC. Their argument could be strengthened by testing if overexpression of dbl-1 is able to rescue the nhsf-1 phenotypes and if dbl-1 expression in neurons and TGF-β/BMP activity in intestine (RAD-SMAD reporter) is increased in hsf-1 (sy441 and/or RNAi) mutants at 25ºC.

3. hsf-1 overexpression in neurons is sufficient and necessary to control fat metabolism in peripheral tissues. However, it is not clear if endogenous hsf-1 also functions in the peripheral non-neuronal tissues to mediate the fat remodeling signaling. Is over-expressing hsf-1 in neurons is still able to regulate fat metabolism in hsf-1(sy441) mutants?

4. The authors argue that lipid remodeling caused by nhsf-1 may provide beneficial effect on lifespan at 25ºC. This is a very interesting hypothesis. However, it is not clear if these changes in lipid metabolism are involved in the nhsf-1 induced lifespan extension. It is also possible that nhsf-1 prolongs lifespan through daf-16/FOXO dependent signaling (Douglas et al., 2015), rather than through lipid remodeling. To determine whether lipid remodeling is sufficient to adapt to higher temperatures the authors can test if fat-6/7 and dbl-1 mutations are also able to extend lifespan at 25ºC.

Other points: 

1. In figure 1f, the authors compare nhsf-1 and SCD KO phospholipid ratios at 20°C with levels in the wt at 15 and 20C. Adding wild type PE/PC ratio would make this comparison more informative. Also it would be good to determine whether nhsf-1 mutants still can modulate the lipid desaturation in response temperature changes or if they are the same at 15, 20 and 25C. 

2. I may have missed it, but I could not find in the methods if the RNAseq experiments were actually done in independent triplicate samples (as is common). This information should be included. 

3. In figure 4 the authors attempt to identify neurons responsible for nhsf-1 fat phenotype by restoring tax-2 function in specific subsets of cells; yet cell specific promoters do not fully rescue. Does tax-2 expressed under the endogenous promotor fully rescue? That seems the first logical experiment to compare it to the partial rescues of more cell specific expression expts. Partial rescue may be caused by mosaic expression. 

Minor points

4. I find the use of "nhsf-1" to refer to expression of a transgene confusing as it closely resembles the nomenclature of gene names. Replacing this with something indicating this is expression of hsf-1 transgene in neurons (i.e. "ExHSF-1n") would make this more obvious.

5. In the survival curve (fig 1B) of nhsf-1#2 418 animals are censored. Why is this number so much higher than the wild type?

6. In figure 1d, the "SCD KO" is compared to wild type. Would be more obvious to just give the genotype: fat-6; fat-7 rather than the acronym.

7. Figure 3a uses a color scale with arbitrary min/max values which make the differences seem more striking than they actually are. Would be more appropriate to use a bar diagram with the table values. It would be good to include tax-4 as well in the data. 

8. Sometimes p values are missing from bar diagrams e.g. In figure 3b&c, tax-4 mutants have a higher baseline level of fat-7 expression than wild type. Is this difference significant? Similar Fig 3e wt tax-2, Fig 3f (tax-2 + or - nhsf-1)

9. In table 2-bodipy assay, there appears to be a typo in the neurons tested column. 'tax-4(+) in AWC…' should be tax-2. 

10. In figure S2a, the authors show worm area/size and reproduction phenotypes for nhsf-1, are these suppressed by loss of tax-2/tax-4? 

11. Figure 4h makes the genetic interaction between HSF-1, TAX-2/TAX-4 and DBL-1 quite confusing as the "triple mutant" is statistically identical to both single mutants and to the wild type. 

12. The authors show induction of hsp16.41 in a subset of neurons in the head, are these neurons that also express TAX-2/TAX-4. This seems to be a central point in their model. A double labeling expt would be easy to do to support their model. 

Reviewer #4: Review for Chauve et al.

In this manuscript, the authors described their new findings that neuronal HSF-1 may serve as a thermostat to sense environmental temperature changes and consequently regulate the lipid storage, fat desaturation, and membrane fluidity of peripheral tissues. Moreover, they have identified a subset of neurons as well as the signaling pathway (i.e. TGF-β/BMP) that may be responsible for this regulation. This is the first study to demonstrate that membrane composition and fluidity may be cell non-autonomously regulated by HSF-1 activity in a group of sensory neurons. 

Overall, the findings are novel and of great interest to the field. The mechanistic insights provided here in the manuscript, although still with some loose ends, further enhance the potential impact of this works. The manuscript will be further improved with some relative minor clarifications and modifications listed below:

1. One important conclusion the authors made is that there are more than six tax-2/tax-4 expressing neurons are responsible for the HSF-1 mediated fat remodeling. However, the six neurons that tax-2(694) actually affected (AQR and five others) were never examined. On the other hand, the neurons (ASG, and five others) that are not altered tax-2(694) were actually examined. Unfortunately, inactivation of individual neuron or a combination of 2 or 3 of this set of neurons does not affect the fat phenotypes. Since experiments eliminating all six of this set of neurons were not performed, it is hard to come to the conclusion that all six neurons are involved. It remains possible that eliminating a combination of one or two of the neurons from each set (i.e. the AQR set and the ASG set) is sufficient to completely rescue the phenotype. 

2. In many cases, the authors describe the suppression or rescue of the phenotypes by percentage (both in figure and in the text), which is somewhat confusing. For example, in line 303 the authors state that " the loss of tax-2 function causes an 80% suppression". This is similar to what is presented in Figure 3A. However, in the figure 3 legend, the same result was described as follows "for the control….., the difference between sibling is 37%, this difference is only 10%......". Similarly, this is also the case in Figure 4H-I, and line 503-505. I find it hard to comprehend. It may be a good idea to present the data in a more straightforward way.

3. Also in line 470, the authors claimed that "In the absence of dbl-1 function, tax-2(p671) restores 30% less fat accumulation than in the presence of dbl-1 function." I am not sure how this is calculated, since I do not see data for nhsf-1#2; dbl-1(nk3) double mutants. Moreover, the difference between nhsf-1; dbl-1; tax-2 triple mutants and nhsf-1 singles seems to be not statistically significant (labeled as "ns" in Figure 4H). 

4. Is the experiments for Fig 1D and 1E done at 15°C or 25°C? For Figure 1F, a WT control at 20°C may be needed as a proper control for the mutants.

5. Line 400-402. The authors state that they have tested both ttx-1 and ttx-3 in Fig 3H and Table 2. But I cannot find any data for ttx-3.

---

## [Decision Letter · Decision Letter 2]

1 Sep 2021

Dear Dr Casanueva,

Thank you for submitting your revised Research Article entitled "Neuronal HSF-1 activity coordinates fat desaturation across tissues in Caenorhabditis elegans via TGF-β/BMP signalling" for publication in PLOS Biology. I have now obtained advice from the original reviewers and have discussed their comments with the Academic Editor. 

The reviews are appended below. As you will see, the reviewers feel the revised manuscript is much improved. However, Reviewer 3 has noted some minor issues that need to be addressed, and Reviewer 1 has raised concerns with the specificity of the RNAi experiments performed here. Having discussed Reviewer 1’s comments with the Academic Editor, we think that it would be essential for a revision to discuss the caveats and limitations of the RNAi experiment.

Based on the reviews, we will probably accept this manuscript for publication, provided you satisfactorily address the remaining points raised by the reviewers. **IMPORTANT: Please also make sure to also address the following data and other policy-related requests, included here: 

1) DATA REQUEST: Please provide the underlying data for each figure in your manuscript, including supplemental figures. This can be provided as a supplemental file or by uploading the data to a publicly available repository. **Please make sure to reference the underlying data in each figure legend. For example you can add the sentence "The data underlying this figure can be found in S1_Data". Please find more detail on this request and on our data policy below my signature. 

2) DATA REQUEST: Thank you for uploading the RNA-seq data files from WT and neuro-shf1 to NCBI GEO. Can you please provide me with a reviewer token so that I can access the data and confirm that it complies with our data availability policy? (Apologies if you have already included the token and I somehow missed it). 

We expect to receive your revised manuscript within two weeks. 

*Published Peer Review History*

*Early Version*

Sincerely,

Lucas Smith, Ph.D.,

Associate Editor,

lsmith@plos.org,

PLOS Biology

DATA POLICY:

Fig 1 A-F; Fig 2C-E,G-H,J-K; Fig 3A,C-F,H; Fig 4 B-D,G; Fig 5 C-K; Fig S1A-B; Fig S2; Fig S3A-B; Fig S4A-D; Fig S5A-B,E; Fig S6C; Fig S7A-B; Fig S8

**Please also ensure that figure legends in your manuscript include information on where the underlying data can be found, and ensure your supplemental data file/s has a legend.

**Please ensure that your Data Statement in the submission system accurately describes where your data can be found.

Reviewer remarks:

Reviewer #1: The authors have revised the manuscript extensively and addressed many of the questions from reviewers. However, the key concern on the role of endogenous hsf-1 in neurons, also raised by another reviewer, still remains. The authors used the strain TU3401:sid-1(pk3321)V,uIs69 [pCFJ190(myo-2p::mCherry) + unc-119p::sid-1), which is neuronal RNAi sensitized yet without evidently blocking RNAi from other tissues (in particular, the intestine). Thus it is still possible that endogenous neuronal Hsf plays negligible role. Without more definitive evidence from neuronal specific loss-of-function and rescue of hsf-1, the findings may be limited by the overexpression effect of hsf-1 and interpretation of such limitation and caveats should be at least discussed explicitly in the paper.

Reviewer #2: As I stated in the first review of the manuscript, the findings presented here are solid and of novelty and importance even if the mechanistic understanding is still somewhat patchy. The authors have gone to great length to address the concerns of all reviewers and in light of new data they have refined their model and interpretations, so the manuscript has certainly improved. I have no more concerns.

Reviewer #3: The authors have performed additional experiments that improved the manuscript. The role of hsf-1 and TAX-2/4 neurons as a thermostat at physiological conditions were clarified.

- Although some minor points were not completely addressed due to time time constraints, I think that the key conclusions of this study are now well supported in the revised manuscript

Some remaining issues

The normalized bar diagram as shown in the the rebuttal should replace fig 3a . Also, the the tax-4 data should be included in fig 3A. Maybe this is an oversight since the authors claim in the rebuttal that 3a was replaced? 

Figure 4H and 6 seem a duplicitous .

Reviewer #4: In this revised manuscript, the authors have made a lot of efforts to address the concerns that all four reviewers raised. Most importantly, the model has been modified based on newly obtained data. For example, the roles of TAX-2/TAX-4 neurons in regulating fat remodeling have been further clarified. Unfortunately, due to technical limitations and time constrain, there are several questions remain unanswered. Overall, I think the manuscript in its current form, even with those unanswered questions, still provided sufficient novel findings that warrant its publication.

---

## [Editor Report · Decision Letter 3]

29 Sep 2021

Dear Dr Casanueva,

On behalf of my colleagues and the Academic Editor, Piali Sengupta, I am pleased to say that we can in principle offer to publish your Research Article "Neuronal HSF-1 activity coordinates fat desaturation across tissues in Caenorhabditis elegans via TGF-β/BMP signalling" in PLOS Biology, provided you address any remaining formatting and reporting issues. These will be detailed in an email that will follow this letter and that you will usually receive within 2-3 business days, during which time no action is required from you. Please note that we will not be able to formally accept your manuscript and schedule it for publication until you have made the required changes.

**Important: As you address the formatting and reporting requests, to come, we also ask that you address the following two editorial request in your revised manuscript. 

1) DATA REQUEST: Thank you for providing the underlying data for each of your figures on the zenodo repository and for referencing this data in your figure legends. Please add the link to access this data (https://zenodo.org/record/5524609#.YUx0eS1Q3tE) to your Data Availability Statement and to each figure legend.

2) TITLE REQUEST: After discussion within the team we think that the title should be modified to highlight further the cell non-autonomous thermostat-based mechanism its functional relevance, as we find those aspects of the work particularly interesting. If you agree, we might suggest the following title: 

"Neuronal HSF-1 coordinates the propagation of fat desaturation across tissues to enable adaptation to high temperatures in C. elegans"

If you opt to change the title, as indicated above, we request that you change it in the revised manuscript and alos update the title in the Editorial Manager system. 

PRESS

Sincerely, 

Lucas Smith, Ph.D. 

Senior Editor 

PLOS Biology

lsmith@plos.org